

# Observations and regional modeling of aerosol speciation and size distribution over Africa and Europe

MENUT Laurent[1], Guillaume SIOUR[2], Sylvain MAILLER[1], Florian COUVIDAT[3], and Bertrand BESSAGNET[3]

[1]Laboratoire de Météorologie Dynamique, UMR CNRS 8539, Ecole Polytechnique, Ecole Normale Supérieure, Université P.M.Curie, Ecole Nationale des Ponts et Chaussées, Palaiseau, France
[2]Laboratoire Inter-Universitaire des Systèmes Atmosphériques, UMR CNRS 7583, Université Paris Est Créteil et Université Paris Diderot, Institut Pierre Simon Laplace, Créteil, France
[3]Institut National de l'Environnement Industriel et des Risques, Verneuil en Halatte, 60550, Parc Technologique ALATA, France

*Correspondence to:* Laurent Menut, menut@lmd.polytechnique.fr

**Abstract.** The aerosol speciation and size distribution is modelled during the summer 2013 and over a large area encompassing Africa, Mediterranean and western Europe. The modelled aerosol is compared to available measurements such as the AERONET Aerosol Optical depth (AOD) and Inversion Size Distribution (ASD) and the EMEP network for surface concentrations of $PM_{2.5}$, $PM_{10}$ and inorganic species (nitrate, sulfate and ammonium). The main goal of this study is to quantify

the model ability to realistically model the speciation and size distribution of the aerosol. Results first showed that the long-range transport pathways is well reproduced and mainly constituted by mineral dust: spatial correlation is $\approx 0.9$ for AOD and Angstrom, when temporal correlation show that the day to day variability is more difficult to reproduce. Over Europe, the $PM_{2.5}$ and $PM_{10}$ have a mean temporal correlation of $\approx 0.4$, but a lowest spatial correlation ($\approx 0.25$ and 0.62, respectively), showing that the fine particules are not well localized or transported. Being short-lived species, the uncertainties on meteorol-

ogy and emissions conduct to these lowest scores. However, time series of $PM_{2.5}$ with the speciation show a good agreement between model and measurements, and are useful to discriminate the aerosol composition. Using a classification from the south (Africa) to the North (northern Europe), it is shown that mineral dust relative contributions decreases from 50% to 10%, when nitrate increases from 0% to 20%, all other species species, sulfate, sea salt, ammonium, elemental carbon, primary organic matter, are constant. The secondary organic aerosol contribution is between 10% and 20% with a maximum at the latitude of

the Mediterranean sea (Spanish stations). For inorganic species, it is shown that nitrate, sulfate and ammonium have a mean temporal correlation of 0.25, 0.37 and 0.17, respectively. The spatial correlation is better (0.25, 0.5 and 0.87) showing that the mean values may be biased but the spatial localization of sulfate and ammonium is well reproduced. The size distribution is compared to the AERONET product and it is shown that the model is able to reproduce the main values for the fine and coarse mode. More in detail, for the fine mode, the model overestimates the aerosol mass in Africa and underestimates in Europe.



# 1 Introduction

For the World Health Organisation (WHO), air pollution is a major environmental risk to health and particularly particulate matter (PM). The most health-damaging particles are those with a diameter of 10 microns or less, ($PM_{10}$), which can penetrate and lodge deep inside the lungs. PM is responsible for a loss of life expectancy particularly when we consider long-term exposure to $PM_{2.5}$, (Martinelli et al., 2013). Particles also play a role on the evolution of climate via direct and indirect effects (Stocker et al., 2013). In Europe, PM is still a major problem for regional air quality (AQ) (Guerreiro et al., 2013), and the member states have to take measures to reduce the exposure to comply with EU standards driven by international guidelines and regulations. A fraction of PM exceedances number is due to long range transport of desert dust issued from the Saharan region, (Rea et al., 2015). In the air quality directive 2008/50/EC (European Union, 2008), chemistry transport models (CTM) are often cited as a technique to be used to assess air quality. The added-value of using models for AQ management is summarized as follows in Rouil and Bessagnet (2014) with for instance the possibility to subtract days of PM exceedances due to a Saharan dust outbreaks. These models are also used for a better understanding of the atmospheric composition and the radiative impact of aerosols over Europe and Africa (Helmert et al., 2007; di Sarra et al., 2008; Vogel et al., 2009; Berg et al., 2015).

Even if the models are useful integrated tools, the measurements are the mandatory step to really understand the processes involved in the aerosol life cycle and thus its evolution in term of composition and size distribution. During the last fifteen years, many field experiments and long-term measurements in specific super-sites were conducted. In Europe, Querol et al. (2004) analyzed several ground $PM_{2.5}$ and $PM_{10}$ measurements to estimate the chemical composition of the aerosol. This aerosol speciation was conducted to identify the relative contributions of organic and elementary carbons (OC and EC), mineral dust, marine and secondary inorganic aerosols. Depending on the measurements period and the location of the instruments, they showed the very high variability of the aerosol speciation in Europe. Also over Europe, Putaud et al. (2004) analyzed a large ensemble of surface measurements to estimate the chemical characteristics of aerosol depending on the measurements location (from urban to background sites). In the French Alps, Aymoz et al. (2004) studied the inorganic components of the aerosol during a Saharan dust long-range transport event. In Spain, Escudero et al. (2007) statistically analyzed surface $PM_{10}$ measurements to extract the relative part of mineral dust coming from Africa. Viana et al. (2008) reviewed the several methodologies of chemical speciation determination for the source apportionment. In the eastern Mediterranean basin for summer 2012, Kostenidou et al. (2015) analyzed the aerosol concentrations and their chemical compositions over the eastern Mediterranean. The fine aerosol ($PM_1$) was found to be dominated by organic aerosol and sulfate. From all these studies, and as synthesized in Kulmala et al. (2011) (after the European EUCARII project), one major conclusion is the need to better understand the aerosols speciation and size distribution. This need is also the conclusion of Laj et al. (2009), where they list all existing methods to have better observations about the aerosol's chemical composition.

In the field of aerosol modeling, many developments were recently done to simulate these complex observations. At the global scale, and knowing the importance of the aerosol load and composition on Earth's climate, models were significantly improved and are able to accurately describe the different steps in the aerosol formation using complex schemes for nucleation, condensation and coagulation, Schutgens and Stier (2014). These global models are compared and their strength and weak-





ness are quantified, as, for example, in Huneeus et al. (2011) for the mineral dust emissions, transport and deposition in the framework of the AEROCOM project. However, due to computational cost, the global models have to use a limited number of modes or bins to describe the aerosol distribution. In addition, the validation of simulations is often restricted to datasets well documented over the globe, i.e. surface PM concentrations (without speciation) and Aerosol Optical Depth (AOD) but with a

low spatial resolution. CTMs at the regional scale simulate the same processes but usually with a more accurate descriptions for the processes involved in the aerosols formation and evolution.

At the regional scale, air quality models tend to underestimate PM and the main discrepancies are often attributed to a lack of emissions or difficulties to reproduce stable meteorological conditions during PM episodes (Bessagnet et al. (2015); Solazzo et al. (2012)). The chemistry of secondary organic species and deposition are also a source of uncertainties (Bergström et al.,

2012; Fountoukis et al., 2014) but the size distribution modeling is poorly adressed in the literature.

To go further in the aerosol's composition behavior understanding, it is now necessary to develop more constrained frameworks for the model versus observations comparisons. The goal is to be able to answer new questions such as: (i) what is the chemical composition of the aerosol during its complete life cycle including emissions, transport and deposition? (ii) is it possible to accurately identify the relative contributions of anthropogenic and natural emissions in the aerosol budget? (iii) if

the surface PM surface concentrations and AOD are well modeled, are we sure there are no compensation errors in the chemical composition and radiative properties of the aerosol? To answer these questions, we use the WRF and CHIMERE regional models to simulate June and July 2013, over a large domain encompassing Africa and Europe. This period corresponds to the ADRIMED project presented in Mallet et al. (2016) and was already studied with these numerical tools in Menut et al. (2015a), Menut et al. (2015b) and Mailler et al. (2016). In this study, the analysis is focused on the aerosol size distribution

and its speciation in Africa and Europe.

The observations data and the models used are described in section 2 and section 3. The comparisons between observed and modeled concentrations are presented in section 4 for the aerosol optical depth (AOD) and the Angström exponent, section 5 for the surface concentrations of $PM_{2.5}$ and $PM_{10}$, section 6 for sulfate, nitrate and ammonium and section 7 for the aerosol size distribution (ASD). Conclusions and perspectives are presented in section 8.

**2  Observations**

Two types of observations are used in this study: (i) the surface concentrations of aerosols species with the EMEP network data and (ii) the column integrated aerosol measurements with the AERONET network data, with Aerosols Optical Depth and size distribution. All stations locations are displayed in Figure 1 with the EMEP stations in red and the AERONET stations in blue.

These stations were selected to cover the studied region: the western Europe and Meditteranean sea, with, in addition,

stations in Africa representative of the mineral dust emissions before transport towards Europe.





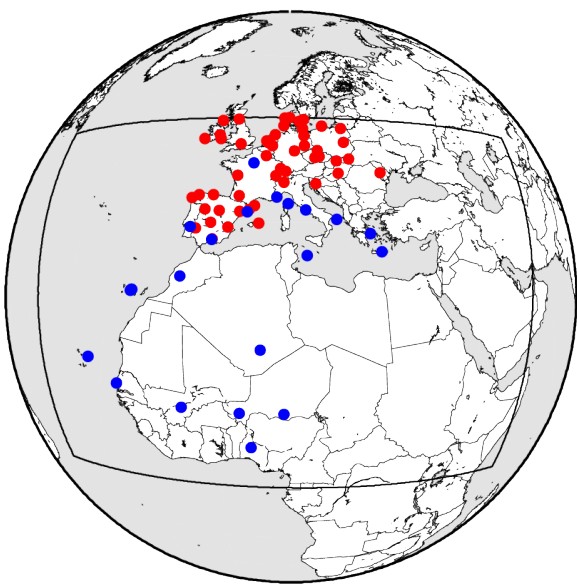

**Figure 1.** *Locations of the measurements stations used in this study. In red the EMEP stations and in blue the AERONET stations. Names and coordinates of these stations are listed in Table 1 and Table 2.*

### 2.1 The EMEP network surface concentrations

For the comparisons between observed and modeled concentrations, the background stations measurements performed during campaigns or in a routine way are used. These measurements are available on the EBAS database (*http://ebas.nilu.no/*) and are used here on a mean daily basis. Only the background stations are used due to the coarse horizontal resolution of the model. Depending on each station, several pollutants are measured: $O_3$, $NO_2$, $SO_2$, $PM_{2.5}$ and $PM_{10}$. For some stations, the inorganic

5  species sulfate, nitrate and ammonium are used. All the stations are listed in Table 1 and located in Figure 1.

### 2.2 The AERONET data

The aerosol optical properties are compared between observations and model using the AERONET (AErosol RObotic NETwork) measurements (Holben et al., 2001). First, the comparison is done using the Aerosol Optical Depth (AOD) measured by the AERONET photometers and for a wavelength of $\lambda$=550nm. The level 2 data are used. Second, the comparison is performed

10  using the Aerosol Size Distribution (ASD) product level 1.5, estimated after inversion of the photometers data as described in Dubovik and King (2000). For each AERONET station used in this study and listed in Table 2, the inversion algorithm provides volume particle size distribution for 15 bins, logarithmically distributed for radius between 0.05 to 15 $\mu$m.



| Site | Altitude | Longitude | Latitude | Site | Altitude | Longitude | Latitude |
|------|----------|-----------|----------|------|----------|-----------|----------|
|      | (m ASL)  | (°)       | (°)      |      | (m ASL)  | (°)       | (°)      |
| Viznar | 1265 | -3.53 | 37.23 | Revin | 390 | 4.63 | 49.90 |
| Barcarrola | 393 | -6.92 | 38.47 | Schmucke | 937 | 10.76 | 50.65 |
| Zarra | 885 | -1.10 | 39.08 | Sniezka | 1603 | 15.73 | 50.73 |
| SanPablo | 917 | -4.34 | 39.54 | Vredepeel | 28 | 5.85 | 51.54 |
| Campisabalos | 1360 | -3.14 | 41.28 | Harwell | 137 | -1.31 | 51.57 |
| Penausende | 985 | -5.86 | 41.28 | Jarczew | 180 | 21.98 | 51.81 |
| ElsTorms | 470 | 0.71 | 41.40 | Valentia | 11 | -10.24 | 51.93 |
| CabodeCreus | 23 | 3.31 | 42.31 | Cabauw | 0 | 4.916 | 51.99 |
| Noya | 683 | -8.92 | 42.72 | Carnsore | 9 | -6.36 | 52.18 |
| OSavinao | 506 | -7.69 | 43.23 | DeZilk | 4 | 4.50 | 52.30 |
| Niembro | 134 | -4.85 | 43.44 | OakPark | 59 | -6.92 | 52.86 |
| Peyrusse | 200 | 0.18 | 43.61 | Neuglobsow | 62 | 13.03 | 53.16 |
| Iskrba | 520 | 14.86 | 45.56 | Kollumerwaard | 1 | 6.27 | 53.33 |
| LeovaII | 166 | 28.28 | 46.48 | DiablaGora | 157 | 22.06 | 54.15 |
| LaTardiere | 133 | -0.75 | 46.65 | Zingst | 1 | 12.73 | 54.43 |
| Payerne | 489 | 6.94 | 46.81 | Leba | 2 | 17.53 | 54.75 |
| K-puszta | 125 | 19.58 | 46.96 | Westerland | 12 | 8.30 | 54.92 |
| Tanikon | 539 | 8.90 | 47.47 | MalinHead | 20 | -7.34 | 55.37 |
| Schauinsland | 1205 | 7.90 | 47.91 | Risoe | 3 | 12.08 | 55.69 |
| Chopok | 2008 | 19.58 | 48.93 | Auchencorth | 260 | -3.24 | 55.79 |
| Starina | 345 | 22.26 | 49.05 | Vavihill | 175 | 13.15 | 56.01 |
| Kosetice | 534 | 15.08 | 49.58 | Ulborg | 10 | 8.43 | 56.28 |
| Svratouch | 737 | 16.05 | 49.73 | Tange | 13 | 9.60 | 56.35 |

**Table 1.** *Names and locations of the EMEP stations used for model comparisons to aerosols surface concentrations. The stations are ordered from South to North. The altitude Above Sea Level (ASL) is indicated since the surface measurements are compared to the first model vertical level.*

## 3 Modeling

For the simulation performed in this study, two regional models are used: (i) the WRF meteorological model calculates the meteorological variables, (ii) the CHIMERE chemistry-transport model calculates the fields concentrations of gaseous and aerosols using the meteorological fields. The horizontal domain is the same for the two models, with a constant horizontal resolution of 60 km × 60 km, as displayed in Figure 1. This domain was selected to be large enough to account for European anthropogenic emissions, African mineral dust emissions and transport of long-lived species across the Mediterranean basin.





| Site | Longitude | Latitude | Code |
|---|---|---|---|
| | $^{o}$ | $^{o}$ | |
| Ilorin | 4.34 | 8.32 | Afr |
| Cinzana | -5.93 | 13.27 | Afr |
| Banizoumbou | 2.66 | 13.54 | Afr |
| ZinderAirport | 8.98 | 13.75 | Afr |
| Dakar | -16.95 | 14.39 | Afr |
| CapoVerde | -22.93 | 16.73 | Afr |
| Tamanrasset | 5.53 | 22.79 | Afr |
| Saada | -8.15 | 31.61 | Afr |
| Izana | -16.49 | 28.31 | Med |
| SantaCruzTenerife | -16.24 | 28.47 | Med |
| LaLaguna | -16.32 | 28.48 | Med |
| ForthCrete | 25.27 | 35.31 | Med |
| Lampedusa | 12.63 | 35.51 | Med |
| Granada | -3.60 | 37.16 | Med |
| Athens | 23.77 | 37.98 | Med |
| Evora | -7.91 | 38.56 | Med |
| LecceUniversity | 18.11 | 40.33 | Med |
| Barcelona | 2.11 | 41.38 | Med |
| RomeTorVergata | 12.64 | 41.84 | Med |
| Bastia | 9.44 | 42.69 | Med |
| Villefranche | 7.32 | 43.68 | Eur |
| Palaiseau | 2.21 | 48.70 | Eur |
| Karlsruhe | 8.428 | 49.093 | Eur |
| Lille | 3.142 | 50.612 | Eur |
| Brussels | 4.350 | 50.783 | Eur |
| Chilbolton | -1.437 | 51.144 | Eur |
| Leipzig | 12.435 | 51.352 | Eur |
| Cabauw | 4.927 | 51.971 | Eur |

**Table 2.** *Names and locations of the AERONET stations used for model comparisons to AOD and ASD data. The stations are ordered from South to North. The altitude ASL are not presented, the measurements being representative of the vertically integrated atmospheric column Above Ground Level (AGL). The three codes are designed to clusterize the results at the end of this study: the classification mainly depends on the latitude of the station to split the domain into three main parts: Africa (for latitude below $\approx 30\,^{o}N$), Med (between latitude $\approx 30\,^{o}N$ and $\approx 45\,^{o}N$), Eur (for latitude up to $\approx 45\,^{o}N$).*





The modeled period ranged from 1st June to 31 July 2013. The results are presented from the 10th June to the 31 July 2013 to
account for a spin-up period.

## 3.1 The WRF meteorological model

The meteorological variables are modeled with the non-hydrostatic WRF regional model in its version 3.6.1, (Skamarock et al.,
2007). The global meteorological analyses from NCEP/GFS are hourly read by WRF using nudging techniques for the main
atmospheric variables (pressure, temperature, humidity, wind). In order to preserve both large-scale circulations and small scale
gradients and variability, the 'spectral nudging' was selected. This nudging was evaluated in regional models, as presented in
Von Storch et al. (2000). In this study, the spectral nudging was selected to be applied for the large-scale dynamics (wave
numbers less than 3 in latitude and longitude, for wind, temperature and humidity and only above 850 hPa). This configuration
allows the regional model to create its own structures within the boundary layer but makes sure it follows the large scale
meteorological fields.

The model is used with 28 vertical levels from the surface to 50 hPa. The Single Moment-5 class microphysics scheme is
used, allowing for mixed phase processes and super cooled water, (Hong et al., 2004). The radiation scheme is RRTMG scheme
with the MCICA method of random cloud overlap, (Mlawer et al., 1997). The surface layer scheme is based on Monin-Obukhov
with Carslon-Boland viscous sub-layer. The surface physics is calculated using the Noah Land Surface Model scheme with
four soil temperature and moisture layers, (Chen and Dudhia, 2001). The planetary boundary layer physics is processed using
the Yonsei University scheme, (Hong et al., 2006) and the cumulus parameterization uses the ensemble scheme of Grell and
Dévényi (2002). The aerosol direct effect is taken into account using the Tegen et al. (1997) climatology.

## 3.2 The CHIMERE chemistry-transport model

### 3.2.1 General overview

CHIMERE is a chemistry-transport model allowing the simulation of concentrations fields of gaseous and aerosols species at a
regional scale. It is an off-line model, driven by pre-calculated meteorological fields. In this study, the version fully described
in Menut et al. (2013a) is used. If the simulation is performed with the same horizontal domain, the 28 vertical levels of the
WRF simulations are projected onto 20 levels from the surface up to 200 hPa for CHIMERE.

The chemical evolution of gaseous species is calculated using the MELCHIOR2 scheme. The photolysis rates are explicitly
calculated using the FastJX radiation module (version 7.0b), (Wild et al., 2000; Bian et al., 2002). The modeled AOD is
calculated by FastJX for the 600nm wavelength over the whole atmospheric column. A complete analysis of the improvement
obtained in the model with this on-line calculation is fully described in Mailler et al. (2016). At the boundaries of the domain,
climatologies from global model simulations are used. In this study, outputs from LMDz-INCA (Szopa et al., 2009) are used
for all gaseous and aerosols species, except for mineral dust where the simulations from the GOCART model are used (Ginoux
et al., 2001).





### 3.2.2 The modelled aerosols

30 The aerosols are modeled using the scheme developed by Bessagnet et al. (2004). This module takes into account sulfate, nitrate, ammonium, primary organic matter (POM) and elemental carbon (EC), secondary organic aerosols (SOA), sea salt, dust and water. The aerosol size is represented using ten bins, from 40 nm to 40 $\mu$m, in mean mass median diameter (MMMD). The aerosol life cycle is completely represented with nucleation of sulfuric acid, coagulation, absorption, wet and dry deposition and scavenging. The scavenging is represented by in-cloud and sub-cloud scavenging.

The aerosol model species and their characteristics are displayed in Table 3. It consists in ten different types of aerosols, some being a compound of several aerosol species.

| Model Species | Origin | Description | Density $\rho_p$ |
|---|---|---|---|
| PPM | anth | Primary Particulate Matter | 1.50 |
| DUST | mineral | Mineral dust | 2.65 |
| EC | anth | Elemental Carbon | 1.50 |
| POM | anth | Particulate Organic Matter | 1.50 |
| SALT | bio | Sea salts | 2.10 |
| SOA | bio/anth | Sec. Organic Aerosols | 1.50 |
| $SO_4$ | anth | Equiv. Sulfate | 1.84 |
| $NO_3$ | anth | Equiv. Nitrate | 1.70 |
| $NH_4$ | anth | Equiv. Ammonium | 1.70 |
| WATER | - | Water | 1.00 |

**Table 3.** *Properties of the modelled aerosol species. The density $\rho_p$ is expressed as value $\times 10^3$ kg m$^{-3}$.*

5 The inorganic part constitutes the major part of the particulate matter in the fine mode (for $D_p < 2.5$ $\mu$m). To determine the gas-particle partitioning of these semi-volatile species, the ISORROPIA model is used (Nenes et al., 1998).

In the model, some processes are certainly roughly or not well represented. For the analysis, it is necessary to consider these approximations. This is the case for the formation of the coarse nitrate aerosol. Coarse nitrate is the result of chemical reaction of nitric acid with mineral dust and sea salt. This process and its impact on the European PM$_{10}$ surface concentrations was 10 studied in a previous version of CHIMERE in Hodzic et al. (2006). In this current version, this process is not yet implemented, due to missing information on the calcium carbonate mass. Thus, the modeled nitrate could be underestimated compared to measurements. Moreover, the formation of SOA formation from Semi Volatile Organic Compound is not represented in this CHIMERE version, since the emission inventories are not mature enough to account for this kind of emissions.



## 3.3 Emissions

Emissions are the only source in the atmospheric composition system, and, thus, represent a large part of the uncertainty in the modeled atmospheric concentrations. This uncertainty is related to the emitted mass flux itself (for gases and aerosol) but also to the size distribution for the modeled aerosol. In this model version, all kind of anthropogenic and natural sources are taken into account on an hourly basis.

### 3.3.1 Emissions fluxes calculations

The anthropogenic emissions are estimated using the same methodology as the one described in Menut et al. (2012) but using the global emission database EDGAR-HTAP annual totals as input data. The EDGAR-HTAP project compiled a global emission dataset with annual inventories for $CH_4$, NMVOC, CO,$SO_2$, $NO_x$, $NH_3$,$PM_{10}$, $PM_{2.5}$, EC and OC, at the national or regional scale that are likely to be acceptable for policy makers in each region of the world. This compilation of different official inventories from EMEP, UNFCCC, EPA for USA, GAINS for China and REAS was first gap-filled with global emission data (Janssens-Maenhout et al. (2012)). The version 2 of this emission inventory was available for the year 2010. The 'fine' part of $H_2SO_4$ corresponds to 1% of the $SO_x$ anthropogenic emissions and thus to primary sulfuric acid. These emissions were already used in this region and for this period in Menut et al. (2015a).

The biogenic emissions are calculated using the MEGAN emissions scheme (Guenther et al., 2006) which provides emission fluxes of nitrogen monoxide, isoprene and monoterpenes.

The mineral dust emissions are calculated using new soil and surface databases described in Menut et al. (2013b) and with a spatial extension of potentially emitting areas in Europe as described in Briant et al. (2014). The dust production model used is the one of Alfaro and Gomes (2001).

The sea salt emissions are calculated following the Monahan (1986) parameterization:

$$
\begin{aligned}
\frac{dF}{dr} &= 1.373 \; u_{10}^{3.41} \; r^{-3} \left(1 + 0.057 \; r^{1.05}\right) \\
&\times \; 10^{1.19 \, exp(-B^2)} \\
\text{with} \quad & B = \frac{0.38 - log(r)}{0.65}
\end{aligned}
\tag{1}
$$

$F$ is the flux of sea salt particle number in unit "particles $m^{-2}$ $s^{-1}$ $\mu m^{-1}$", $r$ the particle radius in $\mu$m and $u_{10}$ is the wind speed at 10 m (in m $s^{-1}$). This is moderated by the percentage of ocean in the grid cell. This formulation directly creates sea salt emission fluxes into the model bins.

### 3.3.2 Emissions distributions in aerosols bins

The way to distribute the primary emissions into the model bins will have a large impact on the finally modeled aerosols. For all aerosols, the primary emissions are provided with three main modes: fine, coarse and big. For each of these modes, a mean





mass median diameter $D_p$ is defined, with its associated $\sigma$. Depending on the emission type (anthropogenic, dust, sea salt, etc.), these parameters are different and are displayed in Table 4.

| Model | $D_p$ ($\mu$m) + ($\sigma$) | | |
|---|---|---|---|
| species | Fine | Coarse | Big |
| SO4 Sulfate | 0.2 (1.6) | | |
| POM Primary Organic matter | 0.2 (1.6) | 4.0 (1.1) | |
| EC Elemental carbon | 0.2 (1.6) | 4.0 (1.1) | |
| PPM Primary Particulate matter | 0.2 (1.6) | 4.0 (1.1) | |
| SALT Sea salts | | Mo86 | |
| WATER Water | | Mo86 | |
| DUST Mineral dust | 1.5 (1.7) | 6.7 (1.6) | 14.2 (1.5) |

**Table 4.** *Aerosols emissions with the three modes describing their size distribution: fine, coarse and big. The mean mass median diameter (MMMD) $D_p$ is expressed in μm, σ is unitless. "Mo86" refers to the parameterization of Monahan (1986).*

For the anthropogenic emissions, the species $SO_4$, POM, EC and PPM are emitted only in the fine and coarse mode, with MMMD of 0.2 $\mu$m and 4 $\mu$m, respectively. Then, log-normal distributions are applied for these two modes to project the emissions into the model bins, as presented in Figure 2. For the sea salt emissions, the distribution is directly the one proposed by Monahan (1986).

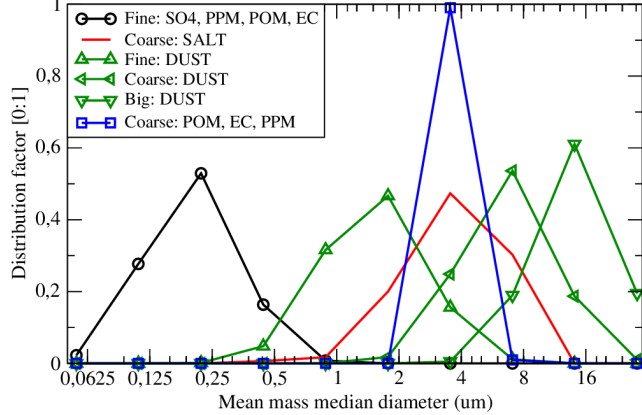

**Figure 2.** *Distribution factors used to project the three aerosols emitted modes on the CHIMERE bins size distribution.*



## 4   Optical properties

In this paper, the first observations *vs* model comparison is done for Aerosol Optical Depth (AOD) and Angström exponent. Correlations are calculated on a daily basis between the AERONET product and the values calculated in CHIMERE using the FastJX module as described in Mailler et al. (2016).

### 4.1   Aerosol Optical Depth

The AOD calculated with CHIMERE does not correspond exactly to the available AERONET data. For the comparison between model and observations, the modelled AOD is interpolated on the AERONET wavelengths. For the region and studied period, the most complete AERONET dataset were found for AOD at $\lambda$=675nm. The CHIMERE AOD useful for the interpolation are for $\lambda$=600nm and 999nm. First, the Angström exponent is estimated as:

$$A(\lambda_1, \lambda_2) = \frac{-log\left(\frac{AOD(\lambda_1)}{AOD(\lambda_2)}\right)}{log\frac{\lambda_1}{\lambda_2}} \qquad (2)$$

where $\lambda_1$ and $\lambda_2$ are two wavelengths and $AOD(\lambda_1)$ and $AOD(\lambda_2)$ the AOD corresponding to these two wavelengths. In case of this study, $\lambda_1$=600nm and $\lambda_2$=999nm with CHIMERE. Then, the interpolated AOD is obtained as:

$$AOD(\lambda_3) = AOD(\lambda_2)exp(-A(\lambda_1, \lambda_2) \times log(\lambda_3/\lambda_2)) \qquad (3)$$

with $\lambda_3$=675nm for the comparison between CHIMERE and AERONET.

For the period from 10th June to 30th July 2013, and for all stations listed in Table 2, number of available data, correlations, Root Mean Square Error (RMSE) and bias are presented in Table 5 for AOD. Generally, the bias is slightly positive for locations close to mineral dust emissions (Banizoumbou, Capo Verde, Dakar and Tamanrasset) and negative for locations far from these sources. This bias ranges from -0.14 (Brussels) to 0.28 (Dakar) and thus represent up to 100% of the AOD value. Compared to the AOD absolute value, the correlation is better: the temporal variability is better captured by the model than the mean average. The temporal variability is primarily explained by the meteorology (for dust emissions, transport and deposition of particles) and these correct correlations show that the model is able to reproduce the observed events of huge aerosol plumes. The absolute value is more difficult to model because of its calculation methodology: the model uses a size distribution with a limited number of bins. Even if this approach is the more realistic to describe the complex behavior of aerosols, it has some limitations: the number of bins and the values of the mean mass median diameter of the primary particles will have a direct impact on all modeled processes (from the emissions to the deposition). The choice of the bins properties has also an impact on the AOD calculation itself: the distribution has to be projected on the extinction efficiency function, characterized by a narrow spread around the measured value. Thus, it is not surprising to have a large variability in AOD modeled values compared to measurements, but it does not mean that the aerosol life cycle is not well represented in the model.





Finally, the last line of Table 5 presents scores for all stations at the same time. $R_s$ represents the correlation between the temporally averaged values of observed and modelled AOD. $R_s$ shows here that the low/high AOD values are very well retrieved by the model, where and when they are observed by AERONET. The mean correlation is +0.3 showing that some stations have low temporal correlations. The mean RMSE is 0.21 the mean bias is 0.02, showing that in average the positive

30  bias (mainly in Africa) compensates the negative one (mainly in Europe).

### 4.2  Angström exponent

In addition, to the aerosol optical depth, the Angström exponent provides a derived information on the size distribution of the aerosols in the vertically integrated atmospheric column. Depending on its value, one can have a first look of the dominant aerosol size in the atmosphere: mainly fine or mainly coarse. For low values, the atmospheric column is mainly composed of coarse particles (mineral dust and sea salt) when for large values the anthropogenic and biogenic contributions dominate.

After a complete screening of the available AERONET data, the most abudant informations are for $A(440, 870)$. In order

to have the same information with CHIMERE, the modelled AODs are first estimated following the interpolation described in equation 2 and for wavelengths $\lambda$=440nm and 870nm. Then, the corresponding Angström exponent is estimated using equation 3.

Results are presented in Table 6. The mean averaged temporal correlation is better than for AOD, with $R$=0.54. This means that the size distribution (fine or coarse) is more accurately modeled than the AOD value itself. The bias (model minus obser-

vations) is large for all stations and negative. More the stations are north and more the bias is important. This means that the model tends to diagnose too low values of Angström exponents, thus atmospheric columns with too much mass of particles in the coarse mode compared to the fine one. The mean spatial correlation $R_s$ is good with $R_s$=0.96. This means that the long range transport and the locations of the aerosol plumes is correctly estimated by the model.

### 4.3  Optical properties maps

In order to have another view of the model results, measured AOD and Angström (AC) are overprinted on maps of these modeled variables in Figure 3. This enables to identify several cases, representative of the diversity of observed situations during this period of June and July 2013. Three days are selected: 18 June, 4 July and 23 July. These days will be used as cases in the following parts of this study. The discussion is focused on the western Africa, Europe and Mediterranean Sea, where the ADRIMED measurements campaign was performed (Mallet et al., 2016).

For the **18 June 2013**, a large dust plume, issued from Africa, reaches the western Europe, leading to large AOD values over France, Benelux and Germany. The plume is not spatially large but with important absolute values of AOD. In Africa,

the model is able to capture the high observed values, up to 0.5. In Europe, the model presents also an intense plume, but the measured values are less important, especially for the 18 June 2013. The corresponding AE map first shows the good agreement between model and measurements. The low values of AE corresponding to the high values of AOD in the plume confirm the mineral dust origin of the aerosol. In addition, AE shows that the low AOD over the Mediterranean are not due to the absence





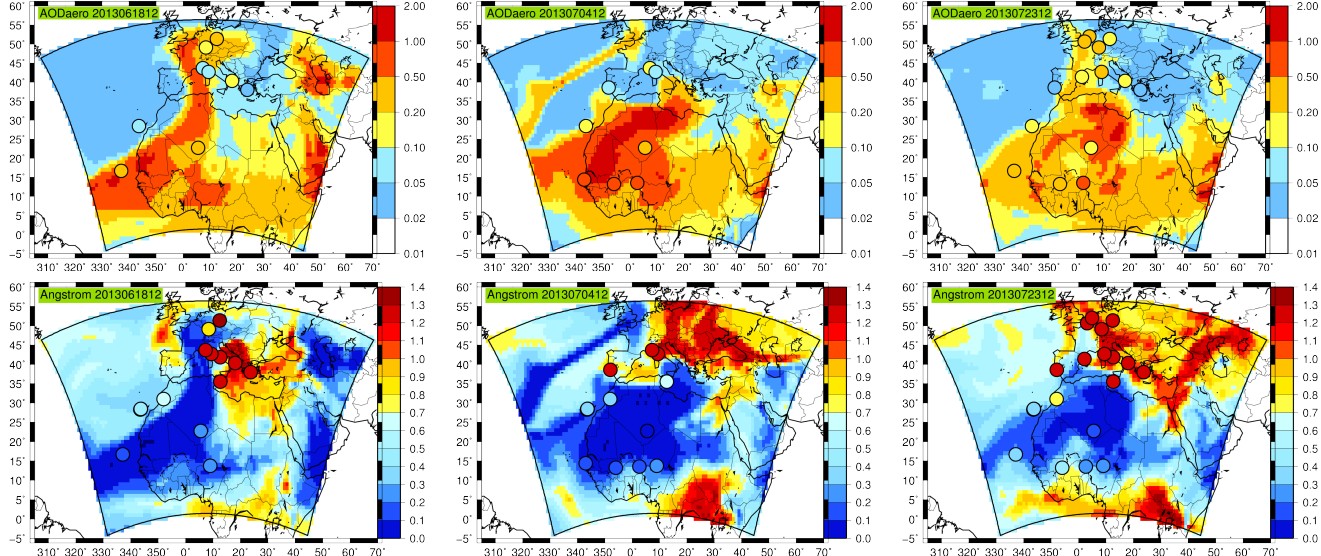

**Figure 3.** *Maps for three different days: (left) 18th june, (middle) 4th july and (right) 23th july and for the AOD (top) and the Angström exponent (bottom). The AERONET measurements are superimposed to the modelled maps in colored circles.*

of aerosols but to anthropogenic and biogenic aerosol with AE values up to one. At the south of the domain, high AE values
are also modeled, showing the African forest fires in Central Africa.

For the **4 July 2013**, a very large area in Africa have high AOD values, up to 0.5. Compared to the measurements, the model overestimates the AOD during the three days. One can also observe a thin plume modeled over the Atlantic sea and flowing until the North of France and the south of United Kingdom. On 5 July, and over the North of France, this plume appears on measurements a little further north than expected in the model simulations. Over western Europe, the AE values increase and
values up to one cover the whole part of this region. Over Africa, AE values are low showing the mineral dust dominance.

For the **23 July 2013**, two plumes are observed from Africa: one to the west and over the Atlantic sea and another one to the Western Europe and over the Mediterranean Sea. The values are less important than for the two other studied days, but the plume has a larger spatial extent and covers the whole western Mediterranean basin. The model is in good agreement with the measurements and the AOD values, between 0.1 and 0.5, are well located by the model. As for the 4 july, the western Europe
is mainly driven by high AE values, corresponding to more fine than coarse aerosol in the whole column: this result is both found for observations and model.



## 5 Surface PM$_{2.5}$ and PM$_{10}$ concentrations

### 5.1 Scores for PM$_{2.5}$ and PM$_{10}$

Comparisons between observed and modeled surface concentrations of PM$_{2.5}$ and PM$_{10}$ are presented in Table 7. Scores are calculated from 10 June to 30 July 2013, leading to a maximum of 51 daily values. The results are presented for the EMEP stations having, at the same time, PM$_{2.5}$ and PM$_{10}$ measurements.

The PM$_{2.5}$ scores show an heterogeneous bias, depending on the location, ranging from -4.35 to +3 $\mu$g m$^{-3}$. Only 5 stations provide measurements for all days. However, except for Payerne (with only 12 days of measurements), all other stations provide more than 40 days on measurements, leading to representative statistics. In general, the correlations are satisfactory and around $\approx$0.5 in average for all stations.

For PM$_{10}$ measurements, only 9 stations out of 25 provide complete times series. The correlation is correct with a large spread in the values: the worst correlation R=-0.11 is calculated in Leova when the best correlation R=0.6 is found at Zarra. For the majority of stations, the model underestimates the concentrations.

More generally, these scores show that the processes leading to fine particles (emissions, chemistry) are better reproduced that the ones at the origin of large particles.

For these comparisons, the scores show that the model is able to reproduce the observed temporal variability. For the aerosol mass, non negligible biases appear with the simulation ($\approx$ 20% of the mass in average), negative or positive, depending of the location. The last line of Table 7 presents the correlation, R$_s$, estimated using the mean averaged values of observed and modeled concentrations. This spatial correlation is better for PM$_{10}$ (R$_s$=0.62) than for PM$_{2.5}$ (R$_s$=0.25). The mean averaged values of correlation are close between PM$_{2.5}$ and PM$_{10}$ with 0.44 and 0.42, respectively. Finally, the averaged bias is larger for PM$_{10}$ (bias=-1.10 $\mu$g m$^{-3}$) than for PM$_{2.5}$ (bias=-0.49), a logical result considering that the aerosol mass is much larger with PM$_{10}$. These scores show that the order of magnitude of ground aerosols concentrations is correctly reproduced.

### 5.2 Time series of PM$_{2.5}$ speciation

Time series of PM$_{2.5}$ are presented to better explain the scores presented in the previous section. For the discussion, six sites are selected. The selection was made independently of the scores found but to be representative of the largest region as possible. The precise location of these sites is displayed in Figure 4 (red symbols). Harwell (United Kingdom) and DiablaGora (Poland) are chosen for the north of Europe, Iskrba (Slovenia) and Schauinsland (Germany) for middle of Europe, Campisabalos and Zarra (Spain) for the south of Europe.

The time series of PM$_{2.5}$ speciation are displayed in Figure 5. The symbols represent the PM$_{2.5}$ EMEP observations. For all sites, the cumulative concentrations until D$_p$ <2.5 $\mu$m of the model species shows a good agreement in term of mass and temporal variability. The important peak of PM$_{2.5}$ observed around the 18 july is well reproduced by the model for stations Harwell, DiablaGora and Iskrba. This peak is overestimated in Schauinsland, mainly due to an overestimation of modeled mineral dust. This peak is mainly due to mineral dust except for Iskrba where this is mostly due a SOA and sulfate peak (mineral dust concentration remains low).





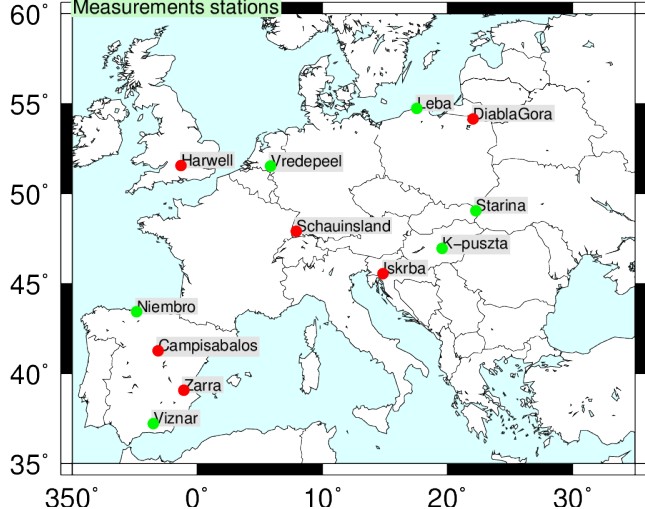

**Figure 4.** *Location of the sites where: (red symbols) surface time series of PM$_{2.5}$ speciations are presented in Figure 5, (green symbols) Time series of PM$_{10}$ for inorganic species are presented in Figure 7.*

The event of 4 July show less important concentrations, meaning that the AOD is more related to long-range transport of aerosols in the troposphere and not to surface concentrations due to local emissions or chemistry. This peak is observed and modeled in Harwell and Campisabalos mainly. At the end of the modeled period, for the event of 23 July, the model is able to reproduce the increase in surface concentrations in Harwell and Campisabalos but failed to estimate the right concentrations in Zarra (overestimation).

The view of the aerosol speciation shows that aerosols peaks, even if they appear at the same period, are not always due to the same chemical species increase.

In order to quantify the relative contribution of each species in the PM$_{2.5}$ concentrations budget, percentages are presented for each EMEP measurements site and in Figure 6. Values are presented for the stations where PM$_{2.5}$ measurements were available. As previously discussed on the PM$_{2.5}$ time series, the chemical composition is dominated by mineral dust and sulfate for all EMEP stations. If the mineral dust and sulfate relative contributions vary a lot (from 10 to 50% for mineral dust and from 20 to 40% for sulfate), the contribution of the other species is less variable: $\approx$ 15% for SOA, $\approx$ 10% for ammonium and less than 10% for the other components.

## 6 Surface inorganic species concentrations

The EMEP network provides surface measurements of nitrate, sulfate and ammonium for aerosol size until 10$\mu$m (PM$_{10}$). This is a good opportunity to evaluate the model capabilities to quantify these chemical species and to determine if the results of the previous sections are not due to error compensations.







**Figure 5.** *Time series of PM$_{2.5}$ (µg m$^{-3}$) with the model aerosol speciation. The colors represent all constituents of the modelled aerosol (for D$_p$ <2.5 µm) and the symbols represent the surface measurements of PM$_{2.5}$.*

From all EMEP stations listed in Table 1, the measurements of these three species are not systematic and regular in time. To quantify the model performance, statistical scores are calculated. The available measurements being different for the three



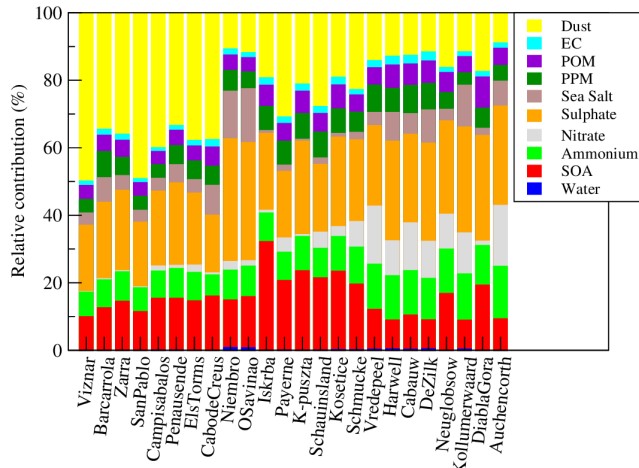

**Figure 6.** *Relative contribution of each chemical species in the budget of the modeled PM$_{2.5}$ surface concentrations for each EMEP station and in average over the period from the 10 June to 30 July 2013.*

species, the results are presented in different tables. The comparison is performed for ammonium, nitrate and sulfate respectively with 21, 25 and 36 stations.

For NH$_4$ comparisons, the results in Table 8 showed a large variability for the correlation. The worst score R=-0.18 is at

Leova, when the best score is at Viznar with R=0.80. The mean absolute values of concentrations are between 0.4 (K-puszta) and 1.6 (DiablaGora) and the RMSE exhibits values with the same order of magnitude, showing a non negligible variability of the error. With values ranging between -0.87 (DiablaGora) and 0.67 (DeZilk), the bias is important and also of the order of magnitude of the mean absolute value. The line 'average' in Table 9 shows that the spatial correlation of NH$_4$ is very low with R$_s$=0.17: this means that the model is not able to retrieve the NH$_4$ plumes of high concentrations where and when they are

observed. The mean averaged bias is +0.16 and represents ≈ 20% of the averaged concentrations, highlighting a non negligible bias with the model for this species.

For SO$_4$, in Table 9, results are better than for ammonium. The correlation R ranges from -0.24 (K-puszta, but this is the only station with a very poor correlation) to 0.78 (OSavinao). The mean values of measured and modeled concentrations are larger than for ammonium and range from ≈ 1 to ≈ 4 $\mu$g m$^{-3}$. The RMSE is satisfactory and never exceeds the half value

of the mean concentration. The bias is scattered ranging from negative (until -0.87 at CabodeCreus) to positive values (until +1.23 at Chopok). The spatial correlation R$_s$=0.5 is better than the one of NH$_4$. The model is more able to retrieve the spatial variability of this pollutant than the temporal variability with the mean averaged correlation of 0.37. The mean bias is very low (+0.05) but the mean RMSE is high (+1.20), showing that the model has the correct order of magnitude for this species but the model variability remains high.

Results for the nitrate are presented in Table 10. The comparison between observation and model is not fair; the model strongly underestimates the observed surface concentrations. In addition, the modeled concentrations temporal variability is



not satisfactorily, with low or negative correlation values. These bad results are mainly due to the missing formation of coarse nitrate. Viznar and Barcarrola illustrates this statement with a strong underestimate of nitrate concentrations, correlated with high simulated dust fraction in the PM$_{10}$ concentrations.

In order to have more information about the temporal variability of these inorganic species concentrations, time series are presented for specific sites where the three species were measured simultaneously with a sufficient number of data. Results are presented in Figure 7 for Leba, Niembro, Starina, Viznar, K-puszta and Vredepeel. These locations are reported in Figure 4. Even if the performances of the model seem poor, these time series show that the order of magnitudes of inorganic species is fairly reproduced (except for nitrate). It means that even if the sources and the chemistry remains uncertain, the inorganic equi-

librium diagnosed using the ISOROPIA module works well to ensure realistic inorganic chemistry and partitioning, whatever the location and the period in summer 2013.

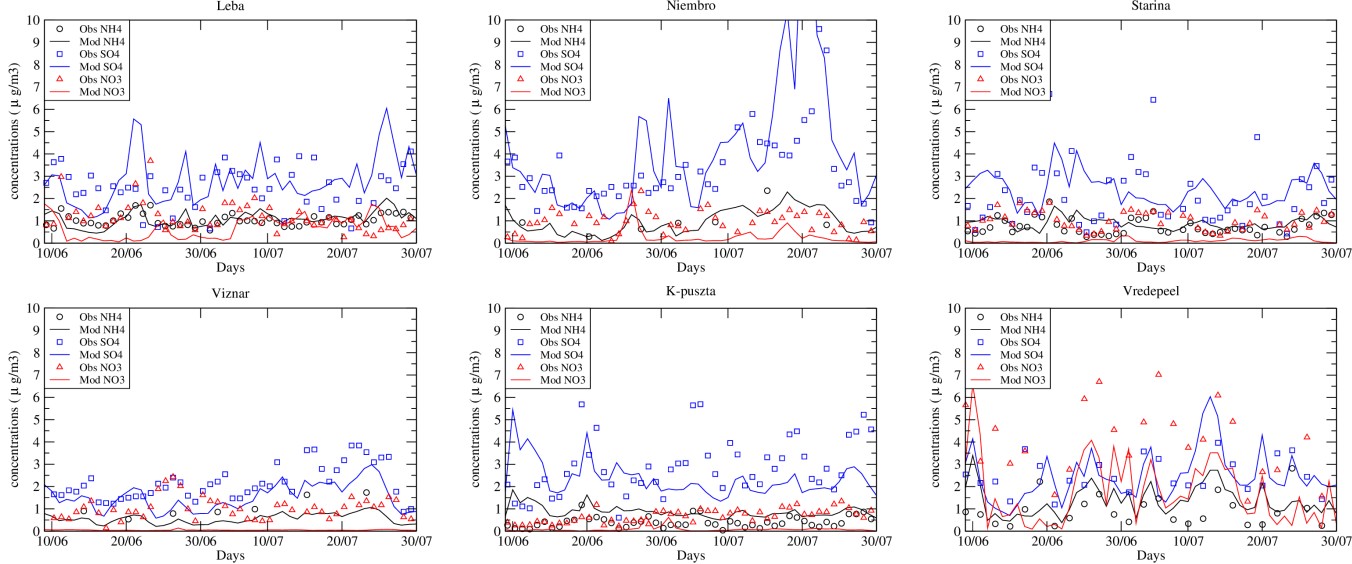

**Figure 7.** *Time series of PM$_{10}$ ($\mu g\ m^{-3}$) for the modeled and measured surface inorganic species.*

Another analysis of the results is presented in Figure 8. The three rows of figures correspond to the three days of 18 June, 4 July and 23 July 2013. The three columns are for sulfate, nitrate and ammonium. For each map, the modeled surface concentrations are expressed in $\mu g\ m^{-3}$ over the whole simulation domain. Since the measurements are restricted to Europe with

the EMEP measurements, a zoom is done to focus on Europe. For each time and each pollutants, the corresponding observed ground concentrations is superimposed as colored circles on the map.

For sulfate, and for the three selected days, the surface concentrations are higher than for nitrate and ammonium, as already discussed in the previous section both for observations and modeling. The most important modeled concentrations are found over the seas (Mediterranean Sea and English Channel). Over land in Europe, the concentrations remain low and the model

5  reproduces well the observed concentrations. Some peaks corresponding to advected plumes are observed and also well mod-



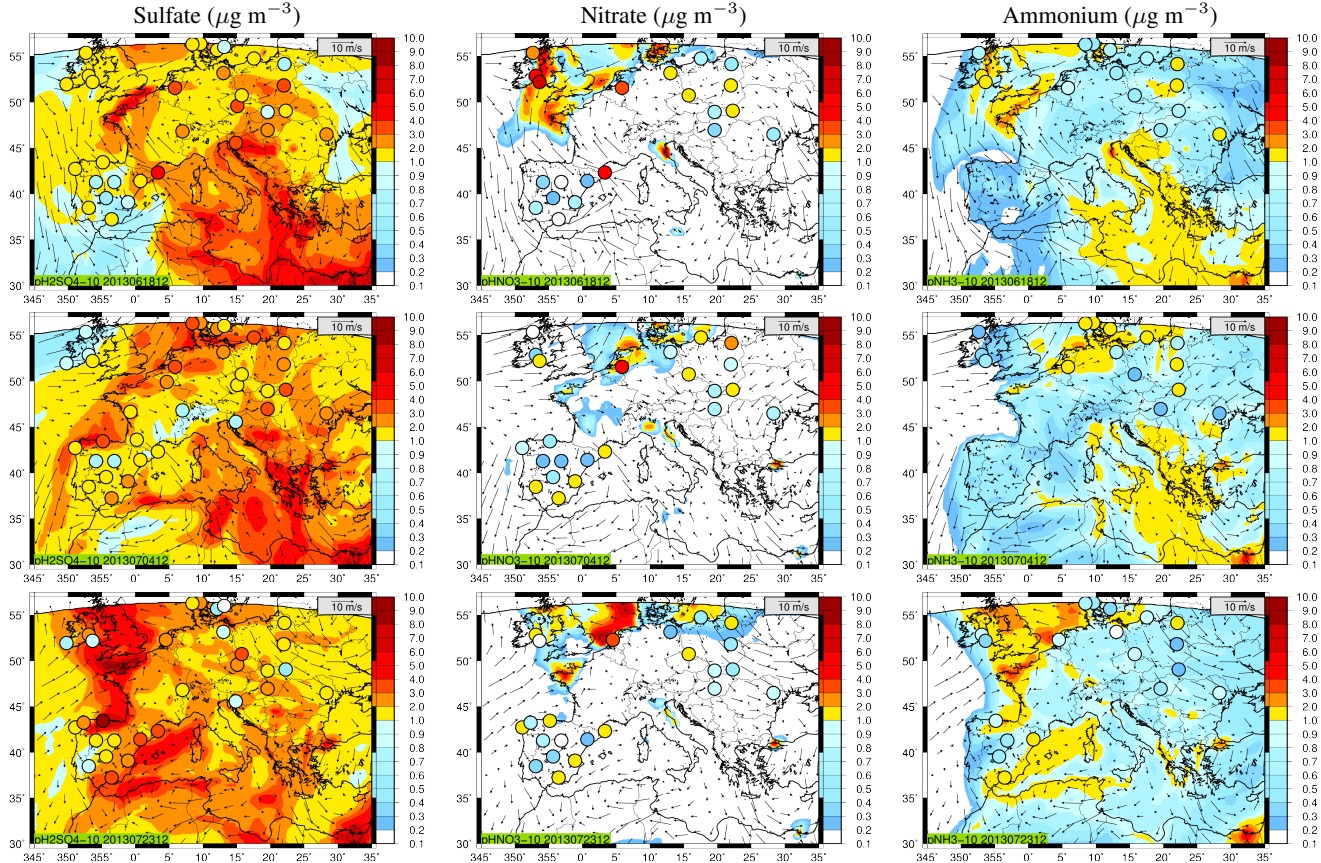

**Figure 8.** *Maps of Sulfate, Nitrate and Ammonium ($\mu$g m$^{-3}$) for the 18 June, 4 and 23 July 2013. A zoom is done over Western Europe where EMEP surface measurements (superimposed to the model) are available. All concentrations values lower than 0.2 $\mu$g m$^{-3}$ are considered as non significant and are not colored. The 10m (above ground level) wind speed is superimposed as vectors.*

eled as in Benelux and Italy (18 June), North of Spain (7 July and 23 July). For this species, the model is able to reproduce the largest spatial patterns with the correct order of magnitude of the concentrations.

For nitrate, the modeled concentrations are low and mainly concentrated in the English Channel. This effectively corresponds to the largest measured values as in the western United Kingdom (18 June), Benelux (7 July and 23 July). The NO$_x$ shipping emissions are responsible for the formation of nitrate favored by mild, humid conditions and low deposition over the Channel. For all other parts of the modeled domain, the model estimates concentrations below 0.2 $\mu$g m$^{-3}$ when the observations ranged between 0.1 and 1 $\mu$g m$^{-3}$, highlighting a systematic underestimation of the model for background values over land.

For ammonium, the modeled background concentrations are higher than for nitrate and ranged from 0.2 to 1 $\mu$g m$^{-3}$. This is in agreement with the observed values and when the highest concentrations are observed, the model simulates a plume close



15   to these areas. Performances on ammonium follow the ones of sulfate, most of the ammonium reacts with sulfuric acid to form

ammonium sulfate salts.

## 7   Aerosol size distribution

In the previous section, the speciation was studied only at the surface using EMEP measurements. An additional way is to

use the AERONET inversions to have aerosol size distribution (ASD) to compare to model results. Two types of comparisons

5  are presented in this section: (i) direct comparison of ASD between model and observations, where and when AERONERT

inversion products are available, (ii) a comparison of fine and coarse modes values to quantify the ability of the model to

estimate the size distribution changes.

### 7.1   ASD speciation

As presented in section 2, the AERONET inversion products provide ASD for 15 bins, following a logarithmic distribution,

ranging from 0.05 to 15 $\mu$m. Using the model aerosol concentrations fields, the column aerosol volume size distribution is

calculated for each model bin $i$ as in Péré et al. (2010):

$$\frac{dV(r_i)}{d\,ln\,r_i} = \sum_{k=1}^{nlevels} \sum_{a=1}^{naero} \frac{m_a(k,r_i) \times \Delta z(k)}{\rho_a \times ln(r_{i,max}/r_{i,min})} \qquad (4)$$

5   where $r_i$ is the mean mass median radius and $r_{i,min}$ and $r_{i,max}$ the boundaries of the $i^{th}$ bin. $m_a(k,r_i)$ is the aerosol mass

concentration (the mass of aerosol in a volume of air, in $\mu$g m$^{-3}$) for the $naero$ modelled aerosols. $\rho_a$ is the aerosol density

(also in $\mu$g m$^{-3}$, the mass of the particle in its own volume). The aerosols densities are fixed per model species and displayed

in Table 3. $\Delta z(k)$ is the model layer thickness (for a total of $nlevels$ levels, here 20 vertical levels). In order to conserve all

model information, the calculation is done on the AERONET bins plus extra bins in the finest and coarsest sizes: 5 bins are

5   added below 0.05 $\mu$m with $r$=0.005, 0.01, 0.02, 0.03 and 0.04 $\mu$m and 3 bins are added after 15 $\mu$m with $r$=20, 30 and 40 $\mu$m.

The model ASD calculation is done independently for each aerosol species in order to have the chemical speciation. All

aerosol ASD are cumulated and are thus directly comparable to the AERONET ASD. Results are presented in Figure 9 for

the three selected periods and for several AERONET stations (chosen to be representative of several locations in the modeled

domain).

10   For model and observations, two main modes are observed: a fine mode with $r \approx 0.1$ $\mu$m and a coarse mode with $r \approx 1$ to

5 $\mu$m. These modes differ a lot between days and locations. On these examples, there is no systematic bias between the model

and the observations regarding the values of the modes radius. A more systematic comparison is presented in the next section.

The speciation is presented for the model and cumulated over all species to have a direct comparison to the AERONET ASD.

For the fine mode, the main modeled species are SOA, sulfate and ammonium. The composition varies a lot from one

15   site to another: in Athens (18 June), SOA and sulfate dominate, when in Evora (23 July) only SOA dominate with a lowest







**Figure 9.** *Comparisons between observed (AERONET) and modeled (CHIMERE) aerosol size distribution for the 18 June, 4 and 23 July 2013. For the model results, the aerosol speciation is displayed with different colors for each species.*

contribution of PPM. For all days and stations, the fine mode is underestimated by the model and exhibits a distribution larger than the AERONET fine mode.

For the coarse mode, the main modeled species is mineral dust. For sites close to this source, the ASD shows a correct order of magnitude (Banizoumbou for the 18 June, Capo Verde for the 4 July). Far from the African dust sources, the mineral dust 20 contribution may be under or overestimated by a factor of two (Evora for the 18 June, Barcelona for the 23 July). The best





comparisons are obtained when the measured coarse mode is centered on $r \approx 2~\mu$m, as, for example, in Banizoumbou (18 June), Izana and Santa Cruz Tenerife (23 July).

## 7.2  ASD fine and coarse modes

In order to have a global view of the model capability to estimate the aerosol size distribution, a simple calculation of these dis-
25    tribution characteristics is done for all sites and hours where AERONET measurements are available. An example is displayed in Figure 10. Most of the AERONET ASD exhibit a two modes distributions, with a "fine" and a "coarse" mode. This is due to the AERONET inversion methodology itself, searching for a local minimum of $dV/dln(r)$ between 0.439 to 0.992 $\mu$m for the aerosol radius. The same analysis is done for the modeled ASD. From these two local minimum values, the local maxima are quantified for the "fine" and "coarse" mode.

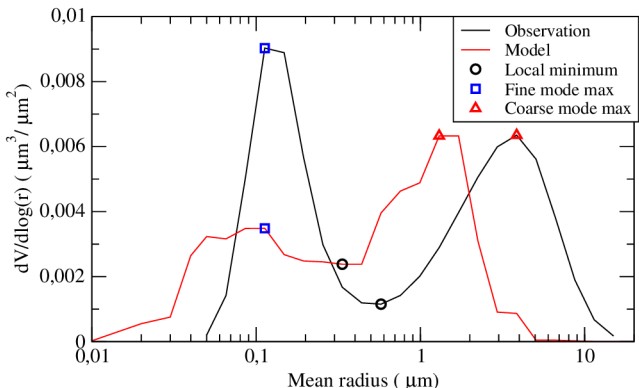

**Figure 10.** *Method for the local mimima and maxima values estimation. This example corresponds to the ASD for Athens, 23 july 2013, 14:00 UTC.*

30    The values of radius are compared between the model and the observations in Figure 11. Since the radius in the size distribution is estimated using a logarithmic progression, the results are also presented using a logarithmic scale. For the observed and modeled distributions, the bins are discretized: this explains the few number of points on the scatter-plot, even if numerous data were analyzed.

The results are classified with three categories: "Africa", "Europe" and "Mediterranean". This classification is related to the stations location (the latitude as explained in Table 2) and enables to see if any systematic trends appear. The results show a large variability of the differences between model and observations, both for the "fine" and "coarse" modes.

For the two modes, this scatter-plot first shows that the variability is larger in the observations than in the model: for one observed specific radius, the model found 3 to 4 different radius, when for one modeled radius, 5 to 6 different radius are found
5    in the observations.

For the "fine" mode and for the stations denoted "Africa", the model overestimates the radius by a factor of two: for the largest occurrences of radius values, when the observations are around $r \approx 0.1~\mu$m, the corresponding model value is $r \approx 0.2$





- 0.3 $\mu$m. For the "Mediterranean" stations, there is a large spread between model and observations but no systematic bias: the fine mode is correctly modeled with $r \approx 0.1$ $\mu$m. For "Europe" stations, the trend is different and a systematic bias appears: in this case, the model underestimates the observed radius by a factor of two.

For the "coarse" mode, the same behavior is observed than for the "fine" mode. A large spread is observed between observations and model, but with well marked trends, depending on the stations location. When the radius is overestimated in Africa, it is well retrieved for Mediterranean stations and underestimated in Europe.

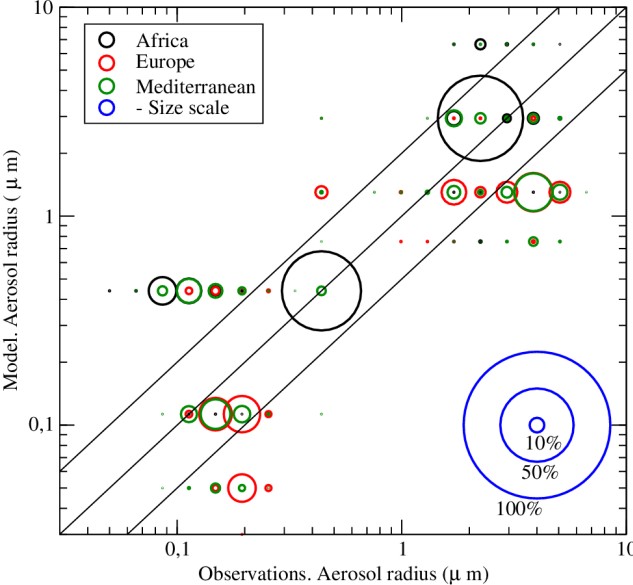

**Figure 11.** *Scatter-plot of the radius found in observations and model for the fine and coarse modes. The width of each symbol represents the occurrence for each obs/model value (normalized with the highest value for each mode "fine" and "coarse" and each location). The blue circles represent the scale for the results, with examples for sizes representing 10, 50 and 100%.*

Another way to quantify the differences between the observed and modeled modes is to integrate the $dV/dlog(r)$ values for the observations and the model, and independently for the "fine" and "coarse" modes. The modes are split considering a constant radius of $r=0.5$ $\mu$m. This choice of a constant value is done to avoid the bias observed in the radius retrieval presented in Figure 11.

Results for this comparison of integrated values are presented in Figure 12. For the fine mode, the cumulated mass of aerosol shows a clear tendency between the three regions: the model overestimates the concentrations in Africa, slightly underestimates the aerosol load in Mediterranean and clearly underestimate the values in Europe.

The results are less marked for the coarse mode but follow the same tendency. In addition, the spread of the cumulated mass is larger than for the fine mode. Over Africa, the model overestimates the aerosol mass, and this concerns high mass values. On the other hand, The model tends to underestimate the mass in the Mediterranean and this corresponds to low mass values.



5   Over Europe, the model underestimates the low mass values, but overestimates the highest mass values. Clearly, the case of
    the Mediterranean stations corresponds to a mixture of anthropogenic and biogenic aerosols (mainly emitted in Europe) and
    mineral dust aerosol (mainly emitted in Africa).

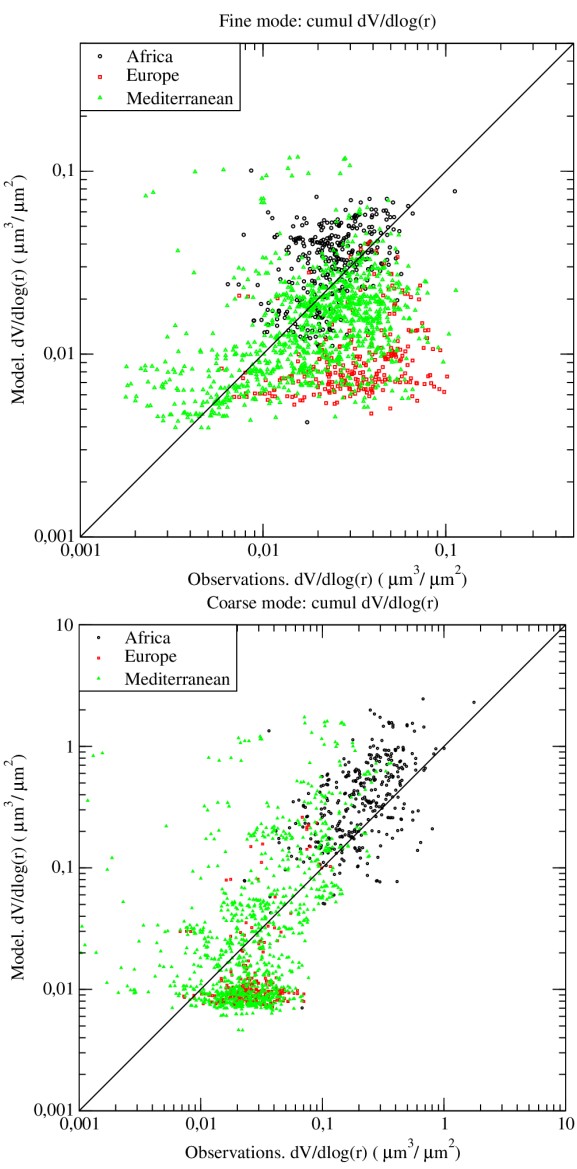

**Figure 12.** *Scatter-plots for comparisons between the observations and the model for the aerosol size distribution.*





## 8 Conclusions

The knowledge of the aerosol composition and size distribution is a scientific challenge for a better understanding of the aerosol life cycle and to improve our understanding of the aerosol impact on health and climate. This is also necessary if we want to split the relative contribution of anthropogenic and biogenic parts in the aerosol to be able to adapt and have more efficient rules in air quality legislation.

This modeling study presents the analysis of a simulation performed with the WRF and CHIMERE models, over a large region including Africa, Mediterranean and western Europe. The simulation was performed for the two months of June and July 2013 and includes all aerosol sources and chemical types. In order to estimate the model accuracy, the AOD and Angström exponent (AE) are compared to the AERONET photometers measurements. For AOD, it is shown that the correlation varies a lot from south (Africa with high correlations) to north (Europe, with low correlations) with a mean averaged value of 0.3. The spatial correlation is better, 0.9, and showed that if the events are not temporally well modelled, the large spatial structures of dense plumes is well estimated by the model. This is confirmed by the good scores with the AE, showing that the origin of the air masses and thus the relative abundance of fine/coarse aerosol is correctly retrieved by the model (spatial correlation of 0.96). The $PM_{2.5}$ and $PM_{10}$ surface concentrations are compared to the EMEP network measurements. A mean averaged correlation of 0.42 and 0.44 is found, with negative biases of -0.49 and -1.10 $\mu g\ m^{-3}$.

To go further in the analysis, several additional measurements are added to this observations versus model comparison. First, this study takes advantages of the availability of surface measurements of inorganic chemical species such as nitrate, sulfate and ammonium. The equivalent species are modelled with CHIMERE and it is shown that the mean averaged correlation is 0.25, 0.37 and 0.17, for these three species respectively. The spatial correlation is different and is 0.25, 0.5 and 0.87, respectively. This shows that if some bias remain in the modeling of these species, the spatial localization of sulfate and ammonium is well captured by the model. The modeling of the nitrate is the weak point for these inorganic species, certainly due to missing sources and processes such as the calculation of coarse nitrate. Second, we take advantage of the AERONET inversion products to estimate the model capability to retrieve the aerosol size distribution over this large region. It is shown that the two main observed modes are well estimated: in Africa, the model is able to correctly estimate the observed radius of the AERONET distribution, when a largest variability is diagnosed in the Mediterranean and Europe. In mass, the aerosol's fine mode is overestimated in Africa, but underestimated in Europe. The Mediterranean having an aerosol being a mix between African sources (mainly mineral dust), local sea salt and European sources, the modelled mass in the fine mode exhibits a large variability compared to the measurements. Results in mass are better for the coarse mode, but always with a slight model overestimation in Africa and a model underestimation in Europe.

This study shows that the chemistry-transport model CHIMERE is able to reproduce the observed variability of the aerosol composition and transport over several regions as Africa, Mediterranean and Europe. By splitting the analysis in term of chemical composition, it is shown that the scores obtained for $PM_{2.5}$ and $PM_{10}$ are not due to model errors compensation, the order of magnitude and time variability of inorganic species being correctly reproduced. The next step will be to reduce the



uncertainties on: (i) the mineral dust emissions in Africa, representing a large part of the model error after long-range transport from Africa to Europe, (ii) the sources and chemistry of nitrate.

*Acknowledgements.* INERIS is funded by the French Ministry in charge of Ecology. The EBAS database has largely been funded by the CLRTAP-EMEP programme, AMAP and by NILU internal resources. Specific developments have been possible due to projects like EU-SAAR (EBAS web interface), EBAS-Online (upgrading of database platform) and HTAP (import and export routines to build a secondary

5    repository for in support of www.htap.org. A large number of specific projects have supported development of data and metadata reporting schemes in dialog with data providers (CREATE, ACTRIS and others). For a complete list of programmes and projects for which EBAS serves as a database, please consult the information box in the Framework filter of the web interface.



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





| Site | $N$ | $AOD$ | | $R_t$ | RMSE | bias |
|---|---|---|---|---|---|---|
| | | Obs | Mod | | | |
| Ilorin | 23 | 0.31 | 0.40 | 0.10 | 0.20 | 0.08 |
| Cinzana | 52 | 0.50 | 0.51 | 0.58 | 0.25 | 0.01 |
| Banizoumbou | 53 | 0.47 | 0.47 | 0.72 | 0.21 | 0.00 |
| ZinderAirport | 55 | 0.45 | 0.65 | 0.69 | 0.38 | 0.20 |
| Dakar | 44 | 0.54 | 0.81 | 0.57 | 0.42 | 0.28 |
| CapoVerde | 41 | 0.47 | 0.67 | 0.48 | 0.35 | 0.19 |
| Tamanrasset | 60 | 0.36 | 0.42 | 0.01 | 0.32 | 0.06 |
| Izana | 60 | 0.06 | 0.21 | 0.44 | 0.37 | 0.16 |
| SantaCruzTenerife | 59 | 0.13 | 0.21 | 0.48 | 0.33 | 0.08 |
| LaLaguna | 54 | 0.12 | 0.23 | 0.48 | 0.35 | 0.10 |
| Saada | 58 | 0.22 | 0.40 | 0.32 | 0.57 | 0.18 |
| ForthCrete | 40 | 0.10 | 0.09 | 0.54 | 0.06 | -0.01 |
| Lampedusa | 43 | 0.15 | 0.25 | 0.63 | 0.26 | 0.10 |
| Granada | 17 | 0.10 | 0.09 | 0.76 | 0.06 | -0.01 |
| Athens | 47 | 0.12 | 0.10 | 0.47 | 0.08 | -0.02 |
| Evora | 56 | 0.09 | 0.06 | 0.25 | 0.08 | -0.03 |
| LecceUniversity | 46 | 0.12 | 0.10 | 0.17 | 0.10 | -0.03 |
| Barcelona | 49 | 0.15 | 0.10 | 0.30 | 0.09 | -0.04 |
| RomeTorVergata | 57 | 0.14 | 0.10 | 0.12 | 0.09 | -0.04 |
| Bastia | 52 | 0.14 | 0.10 | 0.14 | 0.11 | -0.04 |
| Villefranche | 37 | 0.13 | 0.09 | 0.09 | 0.13 | -0.04 |
| Palaiseau | 44 | 0.16 | 0.07 | 0.04 | 0.13 | -0.09 |
| Karlsruhe | 39 | 0.15 | 0.10 | 0.04 | 0.13 | -0.05 |
| Lille | 35 | 0.19 | 0.06 | 0.03 | 0.16 | -0.12 |
| Brussels | 30 | 0.19 | 0.06 | -0.14 | 0.18 | -0.14 |
| Chilbolton | 30 | 0.16 | 0.05 | -0.09 | 0.15 | -0.11 |
| Leipzig | 39 | 0.15 | 0.08 | 0.23 | 0.13 | -0.08 |
| Cabauw | 35 | 0.16 | 0.05 | -0.06 | 0.13 | -0.10 |
| Average | | $R_s$=0.90 | | 0.30 | 0.21 | 0.02 |

**Table 5.** *Scores for the comparisons between observations (AERONET) and model (CHIMERE) for the Aerosol Optical Depth (AOD). Results are presented with $N$ the number of daily mean available measurements for the period from 10th June to 30th July 2013, the temporal correlation ($R_t$), the Root Mean Squared Error (RMSE) and the bias (model minus observations). The last line 'average' represents the spatial correlation $R_s$ between the mean observed and modeled values, and the mean averaged values of temporal correlation, RMSE and bias.*





| Site | $N$ | Angström | | $R_t$ | RMSE | bias |
|---|---|---|---|---|---|---|
| | | Obs | Mod | | | |
| Ilorin | 21 | 0.79 | 0.55 | 0.56 | 0.40 | -0.25 |
| Cinzana | 44 | 0.27 | 0.29 | 0.51 | 0.19 | 0.02 |
| Banizoumbou | 45 | 0.28 | 0.34 | 0.70 | 0.16 | 0.06 |
| ZinderAirport | 46 | 0.32 | 0.26 | 0.71 | 0.16 | -0.07 |
| Dakar | 44 | 0.26 | 0.09 | 0.67 | 0.22 | -0.17 |
| CapoVerde | 36 | 0.17 | 0.09 | 0.72 | 0.11 | -0.09 |
| Tamanrasset | 51 | 0.16 | 0.08 | 0.57 | 0.11 | -0.08 |
| Izana | 51 | 0.61 | 0.32 | 0.75 | 0.38 | -0.29 |
| SantaCruzTenerife | 50 | 0.67 | 0.32 | 0.51 | 0.48 | -0.35 |
| LaLaguna | 46 | 0.60 | 0.30 | 0.50 | 0.44 | -0.30 |
| Saada | 49 | 0.37 | 0.26 | 0.63 | 0.22 | -0.10 |
| ForthCrete | 34 | 1.31 | 0.86 | 0.65 | 0.52 | -0.45 |
| Lampedusa | 43 | 1.17 | 0.64 | 0.80 | 0.61 | -0.53 |
| Granada | 8 | 0.81 | 0.52 | 0.95 | 0.32 | -0.29 |
| Athens | 38 | 1.61 | 0.97 | 0.75 | 0.68 | -0.64 |
| Evora | 49 | 1.31 | 0.70 | 0.32 | 0.68 | -0.61 |
| LecceUniversity | 46 | 1.59 | 1.11 | 0.72 | 0.54 | -0.48 |
| Barcelona | 42 | 1.49 | 0.72 | 0.23 | 0.82 | -0.76 |
| RomeTorVergata | 49 | 1.54 | 0.97 | 0.72 | 0.63 | -0.57 |
| Bastia | 44 | 1.53 | 1.02 | 0.59 | 0.59 | -0.51 |
| Villefranche | 33 | 1.56 | 0.93 | 0.69 | 0.67 | -0.62 |
| Palaiseau | 37 | 1.41 | 0.88 | 0.40 | 0.60 | -0.53 |
| Karlsruhe | 33 | 1.55 | 0.83 | 0.33 | 0.79 | -0.72 |
| Lille | 28 | 1.36 | 0.90 | 0.51 | 0.52 | -0.46 |
| Brussels | 25 | 1.47 | 0.97 | 0.04 | 0.57 | -0.51 |
| Chilbolton | 22 | 1.19 | 0.67 | -0.05 | 0.63 | -0.52 |
| Leipzig | 34 | 1.58 | 0.80 | 0.24 | 0.82 | -0.78 |
| Cabauw | 26 | 1.26 | 0.82 | 0.37 | 0.51 | -0.44 |
| Average | | $R_s$= 0.96 | | 0.54 | 0.48 | -0.39 |

**Table 6.** *Scores for the comparisons between observations (AERONET) and model (CHIMERE) for the Angström exponent. Results are presented with N the number of daily mean available measurements for the period from 10th June to 30th July 2013, the temporal correlation ($R_t$), the Root Mean Squared Error (RMSE) and the bias (model minus observations). The last line 'average' represents the spatial correlation $R_s$ between the mean observed and modeled values, and the mean averaged values of temporal correlation, RMSE and bias.*



| Site | PM$_{2.5}$ | | | | | PM$_{10}$ | | | | | |
|------|---|-----|-----|-------|------|---|-----|-----|-------|------|
| | $N$ | Obs | Mod | $R_t$ | RMSE | bias | $N$ | Obs | Mod | $R_t$ | RMSE | bias |
| Viznar | 46 | 12.48 | 8.13 | 0.44 | 6.07 | -4.35 | 48 | 22.00 | 15.35 | 0.39 | 14.84 | -6.65 |
| Barcarrola | 47 | 9.77 | 7.50 | 0.33 | 5.71 | -2.26 | 50 | 16.96 | 11.78 | 0.13 | 16.50 | -5.18 |
| Zarra | 49 | 7.73 | 8.91 | 0.45 | 4.85 | 1.17 | 50 | 14.82 | 16.77 | 0.60 | 16.63 | 1.95 |
| SanPablo | 51 | 8.00 | 6.40 | 0.48 | 3.53 | -1.60 | 51 | 15.20 | 10.12 | 0.24 | 10.51 | -5.08 |
| Campisabalos | 43 | 9.56 | 7.49 | 0.58 | 4.01 | -2.07 | 45 | 10.98 | 10.78 | 0.38 | 8.35 | -0.20 |
| Penausende | 49 | 6.65 | 6.22 | 0.56 | 2.91 | -0.43 | 50 | 11.06 | 7.82 | 0.38 | 5.80 | -3.24 |
| ElsTorms | 46 | 8.30 | 10.34 | 0.48 | 5.50 | 2.04 | 49 | 14.53 | 19.84 | 0.37 | 21.57 | 5.31 |
| CabodeCreus | 46 | 9.33 | 12.23 | 0.16 | 7.73 | 2.90 | 46 | 18.35 | 31.07 | 0.26 | 42.79 | 12.72 |
| OSavinao | 43 | 10.14 | 9.03 | 0.68 | 3.93 | -1.11 | 43 | 13.26 | 13.14 | 0.53 | 4.42 | -0.12 |
| Niembro | 48 | 8.23 | 10.49 | 0.58 | 5.04 | 2.26 | 48 | 17.02 | 14.55 | 0.58 | 6.53 | -2.47 |
| Iskrba | 51 | 10.82 | 9.31 | 0.47 | 5.47 | -1.51 | 51 | 13.96 | 11.12 | 0.33 | 9.08 | -2.84 |
| Payerne | 12 | 10.77 | 8.48 | 0.47 | 4.47 | -2.28 | 51 | 14.13 | 12.73 | 0.47 | 11.60 | -1.40 |
| Schauinsland | 48 | 9.65 | 9.61 | 0.09 | 7.22 | -0.04 | 49 | 12.35 | 12.21 | 0.15 | 10.97 | -0.14 |
| Kosetice | 25 | 11.52 | 8.50 | 0.44 | 5.55 | -3.02 | 25 | 11.16 | 9.43 | 0.49 | 5.70 | -1.73 |
| Schmucke | 51 | 8.11 | 7.95 | 0.41 | 5.02 | -0.15 | 51 | 11.95 | 9.36 | 0.44 | 7.15 | -2.59 |
| Harwell | 51 | 7.81 | 7.96 | 0.63 | 3.84 | 0.15 | 51 | 13.24 | 9.84 | 0.56 | 6.21 | -3.40 |
| Neuglobsow | 51 | 7.32 | 7.64 | 0.16 | 5.06 | 0.32 | 50 | 11.05 | 8.51 | 0.14 | 6.11 | -2.54 |
| DiablaGora | 50 | 8.22 | 6.24 | 0.52 | 3.52 | -1.98 | 51 | 11.43 | 7.25 | 0.57 | 5.46 | -4.18 |
| Auchencorth | 41 | 5.22 | 7.89 | 0.48 | 3.86 | 2.67 | 2 | 7.00 | 7.89 | 1.00 | 0.99 | 0.89 |
| Average | | R$_s$=0.25 | | 0.44 | 4.91 | -0.49 | | R$_s$=0.62 | | 0.42 | 11.12 | -1.10 |

**Table 7.** *Scores for the comparisons between observations (EMEP) and model (CHIMERE) for PM$_{2.5}$ and PM$_{10}$.. Results are presented with N the number of daily mean available measurements for the period from 10th June to 30th July 2013, the temporal correlation ($R_t$), the Root Mean Squared Error (RMSE) and the bias (model minus observations). The last line 'average' represents the spatial correlation $R_s$ between the mean observed and modeled values, and the mean averaged values of temporal correlation, RMSE and bias.*





| Site | $N$ | NH$_4$ | | R$_t$ | RMSE | bias |
|---|---|---|---|---|---|---|
| | | Obs | Mod | | | |
| Viznar | 7 | 1.06 | 0.57 | 0.80 | 0.57 | -0.49 |
| SanPablo | 7 | 0.55 | 0.46 | 0.18 | 0.23 | -0.09 |
| Campisabalos | 7 | 0.48 | 0.79 | 0.39 | 0.43 | 0.31 |
| ElsTorms | 7 | 0.84 | 0.90 | 0.74 | 0.20 | 0.06 |
| Niembro | 7 | 0.98 | 1.00 | 0.22 | 0.80 | 0.02 |
| LeovaII | 51 | 0.60 | 0.85 | -0.18 | 0.81 | 0.25 |
| K-puszta | 51 | 0.40 | 0.78 | -0.07 | 0.52 | 0.38 |
| Starina | 49 | 0.75 | 0.89 | 0.01 | 0.48 | 0.14 |
| Sniezka | 51 | 0.54 | 0.70 | 0.07 | 0.33 | 0.16 |
| Vredepeel | 26 | 0.88 | 1.40 | 0.10 | 1.25 | 0.52 |
| Jarczew | 46 | 1.13 | 0.83 | 0.27 | 0.52 | -0.31 |
| Carnsore | 51 | 0.54 | 0.72 | 0.12 | 0.72 | 0.18 |
| DeZilk | 25 | 0.64 | 1.31 | 0.76 | 0.98 | 0.67 |
| OakPark | 51 | 0.67 | 0.83 | 0.78 | 0.53 | 0.16 |
| Neuglobsow | 51 | 0.38 | 0.96 | -0.10 | 0.73 | 0.58 |
| DiablaGora | 49 | 1.60 | 0.73 | 0.06 | 1.39 | -0.87 |
| Leba | 51 | 1.03 | 1.01 | 0.42 | 0.38 | -0.02 |
| MalinHead | 44 | 0.47 | 0.70 | 0.59 | 0.48 | 0.22 |
| Risoe | 49 | 0.84 | 1.47 | -0.02 | 1.19 | 0.63 |
| Ulborg | 51 | 0.89 | 1.28 | 0.08 | 0.84 | 0.39 |
| Tange | 51 | 0.98 | 1.38 | 0.06 | 0.89 | 0.40 |
| Average | | R$_s$=0.17 | | 0.25 | 0.68 | 0.16 |

**Table 8.** *Scores for the comparisons between observations (EMEP) and model (CHIMERE) for the NH$_4$ surface concentrations (in µg m$^{-3}$). Results are presented with $N$ the number of daily mean available measurements for the period from 10th June to 30th July 2013, the observed and modeled surface concentrations ('obs' and 'mod'), the temporal correlation ($R_t$), the Root Mean Squared Error (RMSE) and the absolute bias (model minus observations). The last line 'average' represents the spatial correlation $R_s$ between the mean observed and modeled values, and the mean averaged values of temporal correlation, RMSE and bias.*





| Site | $N$ | SO$_4$ | | R$_t$ | RMSE | bias |
|---|---|---|---|---|---|---|
| | | Obs | Mod | | | |
| Viznar | 49 | 2.15 | 1.48 | 0.65 | 0.89 | -0.66 |
| Barcarrola | 50 | 1.84 | 1.45 | 0.73 | 0.84 | -0.40 |
| Zarra | 50 | 2.11 | 1.87 | 0.53 | 0.91 | -0.24 |
| SanPablo | 51 | 1.46 | 1.15 | 0.60 | 0.60 | -0.31 |
| Campisabalos | 45 | 1.26 | 1.67 | 0.48 | 0.81 | 0.41 |
| Penausende | 50 | 1.34 | 1.46 | 0.18 | 0.83 | 0.12 |
| ElsTorms | 49 | 2.19 | 1.95 | 0.45 | 0.95 | -0.25 |
| CabodeCreus | 46 | 2.78 | 1.91 | 0.64 | 1.25 | -0.87 |
| Noya | 50 | 1.86 | 2.31 | 0.56 | 1.47 | 0.45 |
| Niembro | 48 | 3.27 | 3.53 | 0.74 | 2.05 | 0.26 |
| OSavinao | 43 | 2.29 | 3.02 | 0.78 | 1.85 | 0.73 |
| Peyrusse | 15 | 2.55 | 1.95 | 0.47 | 1.31 | -0.60 |
| Iskrba | 47 | 1.88 | 2.00 | 0.68 | 0.81 | 0.13 |
| LeovaII | 51 | 2.20 | 2.38 | 0.02 | 2.25 | 0.18 |
| LaTardiere | 15 | 2.01 | 1.92 | 0.64 | 0.69 | -0.09 |
| Payerne | 51 | 1.77 | 1.66 | 0.44 | 0.77 | -0.11 |
| K-puszta | 51 | 2.89 | 2.07 | -0.24 | 1.74 | -0.82 |
| Chopok | 50 | 1.14 | 2.36 | 0.17 | 1.57 | 1.23 |
| Starina | 49 | 2.18 | 2.38 | 0.02 | 1.57 | 0.21 |
| Kosetice | 51 | 2.50 | 1.67 | 0.27 | 1.52 | -0.83 |
| Revin | 15 | 1.99 | 2.25 | 0.72 | 0.86 | 0.26 |
| Sniezka | 51 | 2.20 | 1.74 | 0.15 | 0.98 | -0.47 |
| Vredepeel | 26 | 2.45 | 2.22 | -0.03 | 1.24 | -0.23 |
| Jarczew | 43 | 2.85 | 2.24 | 0.34 | 1.69 | -0.61 |
| Valentia | 51 | 1.27 | 1.80 | 0.59 | 1.09 | 0.53 |
| Carnsore | 51 | 1.79 | 2.07 | 0.10 | 1.66 | 0.27 |
| DeZilk | 25 | 2.49 | 2.87 | 0.48 | 1.25 | 0.38 |
| OakPark | 51 | 1.54 | 2.05 | 0.67 | 1.45 | 0.51 |
| Neuglobsow | 51 | 1.64 | 2.05 | 0.06 | 1.01 | 0.41 |
| DiablaGora | 51 | 1.21 | 1.92 | 0.08 | 1.08 | 0.71 |
| Leba | 49 | 2.49 | 2.61 | 0.12 | 1.42 | 0.12 |
| MalinHead | 44 | 1.28 | 1.57 | 0.59 | 0.73 | 0.29 |
| Risoe | 51 | 1.85 | 2.11 | 0.10 | 1.08 | 0.27 |
| Vavihill | 44 | 1.10 | 2.08 | 0.33 | 1.16 | 0.98 |
| Ulborg | 51 | 2.31 | 2.23 | 0.09 | 0.93 | -0.09 |
| Tange | 50 | 2.00 | 2.08 | 0.11 | 0.84 | 0.08 |
| Average | | R$_s$=0.50 | | 0.37 | 1.20 | 0.05 |

**Table 9.** *Scores for the comparisons between observations (EMEP) and model (CHIMERE) for the SO$_4$. Results are presented with $N$ the number of daily mean available measurements for the period from 10th June to 30th July 2013, the temporal correlation ($R_t$), the Root Mean Squared Error (RMSE) and the bias (model minus observations). The last line 'average' represents the spatial correlation $R_s$ between the mean observed and modeled values and the mean averaged values of correlation, RMSE and bias.*





| Site | $N$ | $NO_3$ | | $R_t$ | RMSE | bias |
|---|---|---|---|---|---|---|
| | | Obs | Mod | | | |
| Viznar | 49 | 1.02 | 0.05 | 0.17 | 1.08 | -0.97 |
| Barcarrola | 50 | 0.83 | 0.08 | -0.16 | 0.84 | -0.75 |
| Zarra | 50 | 1.39 | 0.07 | 0.11 | 1.41 | -1.32 |
| SanPablo | 51 | 0.54 | 0.05 | -0.10 | 0.60 | -0.49 |
| Campisabalos | 38 | 0.28 | 0.13 | 0.72 | 0.27 | -0.15 |
| Penausende | 50 | 0.64 | 0.12 | -0.02 | 0.61 | -0.51 |
| ElsTorms | 49 | 0.47 | 0.25 | -0.07 | 0.54 | -0.21 |
| CabodeCreus | 46 | 1.40 | 0.10 | 0.09 | 1.46 | -1.29 |
| Noya | 40 | 0.90 | 0.16 | 0.14 | 0.91 | -0.74 |
| Niembro | 46 | 0.97 | 0.24 | 0.41 | 0.86 | -0.73 |
| OSavinao | 41 | 0.74 | 0.18 | 0.28 | 0.62 | -0.56 |
| LeovaII | 51 | 0.71 | 0.05 | 0.41 | 0.79 | -0.66 |
| K-puszta | 51 | 0.69 | 0.09 | -0.04 | 0.69 | -0.60 |
| Chopok | 50 | 0.60 | 0.17 | -0.13 | 0.61 | -0.43 |
| Starina | 49 | 1.01 | 0.11 | 0.04 | 0.99 | -0.90 |
| Sniezka | 51 | 1.54 | 0.32 | 0.17 | 1.38 | -1.22 |
| Vredepeel | 26 | 4.53 | 2.19 | 0.01 | 4.40 | -2.34 |
| Jarczew | 46 | 1.17 | 0.22 | 0.08 | 1.11 | -0.95 |
| Carnsore | 50 | 1.60 | 0.40 | 0.21 | 2.12 | -1.21 |
| DeZilk | 25 | 3.73 | 1.59 | 0.78 | 3.95 | -2.14 |
| OakPark | 51 | 1.30 | 0.55 | 0.71 | 1.14 | -0.75 |
| Neuglobsow | 51 | 0.65 | 0.80 | 0.21 | 1.01 | 0.15 |
| DiablaGora | 50 | 1.28 | 0.24 | -0.01 | 1.35 | -1.04 |
| Leba | 51 | 1.16 | 0.65 | -0.22 | 1.05 | -0.51 |
| MalinHead | 44 | 0.84 | 0.71 | 0.45 | 0.99 | -0.13 |
| Average | | $R_s$=0.87 | | 0.17 | 1.23 | -0.82 |

**Table 10.** *Scores for the comparisons between observations (EMEP) and model (CHIMERE) for the nitrate. Results are presented with $N$ the number of daily mean available measurements for the period from 10th June to 30th July 2013, the temporal correlation ($R_t$), the Root Mean Squared Error (RMSE) and the bias (model minus observations). The last line 'average' represents the spatial correlation $R_s$ between the mean observed and modeled values, and the mean averaged values of correlation, RMSE and bias.*