# Peer review of "Observations and regional modeling of aerosol speciation and size distribution over Africa and Europe"

_Atmospheric Chemistry and Physics, 2016_

## Referee Comment (RC1) · Anonymous Referee #2 · 31 Jul 2016

This paper provides a thorough comparison of the performance of the CHIMERE model against a range of ground-based (EMEP stations) and column integrated (NASA-AERONET) aerosol observations and tests the capacity of the model to reproduce critical properties like chemical composition and size distribution over a vast region (Africa/Europe) with contrasted aerosol sources.

This paper provides a substantial contribution to scientific progress within the scope of ACP (substantial new concepts, ideas, methods, and data). The scientific approach and applied methods are correctly presented and the results discussed in an appropriate and balanced way. The scientific results and conclusions presented in a clear, concise, and well-structured way. I do recommend the publication of this paper after

addressing the few minor comments reported below:

Introduction: Can the authors better justify (in one or two sentences) why they have decided to focus on Europe/Mediterranean and why they have decided to use WRF + CHIMERE? Part of the explanation is provided at the beginning of the modelling section and could be moved in the introduction.

Page 2, line 10: "the size distribution modeling is poorly addressed in the literature". Can the authors justify in one sentence why it is a critical parameter that needs to be better constrained by model?

Figure 1: Why the selected NASA-AERONET stations are mostly located in Africa and Southern Europe and the EMEP stations mostly in the Northern Europe? There are also NASA-AERONET stations in Northern Europe and EMEP stations in Southern Europe (Italy, Greece, Cyprus ...)

Page 5, line 18 : "... and transport of long-lived species across the Mediterranean Basin." What are these long-lived species? Table 2: Can the authors add a column "country" to help the reader to better locate the stations?

Page 7, line 9 to 15 (section 3.1): An exhaustive description of the model is reported here. Is it the first time that all of these features are applied to CHIMERE? If not, can the authors provide some references which have used CHIMERE with these features?

Table 3: I still don't understand what are the chemical species behind the term "primary particulate matter" (PPM). Can you be more specific?

Page 9, line 5 (section 3.3.2.). Can the authors provide a size range for fine/coarse/big?

Table 4: What's about the other chemical species described in Table 3?

Figure 2: This figure provides the % distribution in the different modes for the different chemical species. Since this figure is scaled with relative contributions (%), it is not possible, for instance, to learn what is the fine versus coarse contribution of SO4 (e.g.

what is the contribution of fine SO4 relatively to total SO4)?

Table 5: Can the authors specify what the rationale of the classification of the stations is? (e.g. why this order?).

Page 13, line 17: Can the authors explain why they have selected specifically these 3 days?

Page 17, line12: "... the important peak of PM2.5 observed around the 18 July ...". I cannot see clearly a peak of PM2.5 on 18.07 in Figure 5. Can the authors better point this peak?

Page 19, end of the page, discussion related to NH4. The authors discuss on the model performance for NH4 taken individually while this compound is intimately linked to fine SO4 (and fine NO3 if there is). How the neutralization of SO4 by NH4 is reproduced by the model?

Page 22, line 4. Although we usually find the highest concentrations of fine NO3 close to the English channels, I would assume that it is rather due to thermodynamic equilibrium (cold and humid) rather than high NOx from shipping emissions. Can the authors demonstrate that fine NO3 in this region is mostly due to shipping rather than NOx for the continental (transport) sector? Otherwise I would remain vague about the origin of NOx in the region.

Figure 9: Why the authors did not take daily averages here (which are more representative than a given hour)?
* * *

---

## Referee Comment (RC2) · Anonymous Referee #1 · 2 Sep 2016

This manuscript, entitled "Observations and regional modeling of aerosol speciation and size distribution over Africa and Europe" presents an original analysis of aerosol properties, both modeled by a regional chemistry-transport model and given by surface and column-integrated observations, in terms of speciation and size distribution. Detailed results are about different aerosol species, from both natural and anthropogenic origins, are discussed, showing the performance of this regional model to reproduce the variety of aerosols in this region. The methodology and the results are for most of them (except in Section 7) clearly presented and explained, with discussion on the strengths and weaknesses of the current version of the model. However, several corrections are needed before considering a publication in ACP.

[Figure]

——————————————————— Main comments :

1. Even if the comparisons between the model and observations show for different variables satisfactory scores for CHIMERE, the authors tend to overestimate its performance throughout the manuscript, notably in the following cases: - the correlations for AOD (Table 5, Section 4.1) are very weak for many stations (9 out of 28 have a correlation lower than 0.1). - the optical properties maps (Figure 3) show difficulties to reproduce correctly the Angstrom exponent in Western Europe - in the PM2.5 time series, some observed peaks are missed by the model, which is not explained (for example around 10 July in Schauinsland) I do not really challenge the performance of the model as I am aware of the complexity to compare observations and models, but rather the way it is presented by the authors (for example lines16-18 page 11). The latter should moderate their conclusions, and give possible reasons to explain the difficulties mentioned above.

2. The title does not exactly correspond to the manuscript in terms of variables (Section 4 deals with aerosol optical properties which are not mentioned in the title) and of the domain of study ("Africa and Europe" is not precise enough). I suggest to add in the title optical properties, and to replace "Africa and Europe" by "Western Europe and Northern Africa".

3. The spatial distribution of the stations used here could be a major limitation of the study. Indeed, the EMEP stations cover only northern Europe and Spain (Africa and southern Europe are missing), while the AERONET stations do not include northern Europe. To my knowledge, there are other stations which could be taken into account, which would improve the robustness of the different scores.

4. The analyses on the speciation data are very interesting (notably Figures 5 and 6), but the same kind of diagnostics is missing as far as optical properties are concerned. I suggest to the authors to add this information on speciation for AOD, in order to better understand the differences shown in Figure 3.

———————————————————— Minor comments:

- Abstract lines 6-7: I don't understand why only mineral dust is mentioned in this sentence, while spatial correlation and daily variability concern all aerosols.

- Abstract line 12: "dust relative contribution": Please precise the variable (mass ?).

- Page 3 line 18: what does the ADRIMED project bring to the present study ? Are there any specific observations that could be used here ?

- Page 5 lines17-18: The authors should justify properly that "the domain was selected to be large enough to account for anthropogenic emissions". The latitude of the northern limit of the domain seems to be too low, as northern British Isles, a part of northern Sea, and Scandinavia are not included in the domain.

- Page 7 line 1: "the results are presented from 10th June to 31st July 2013". Is it 30th (as in the following sections) or 31st July ?

- Page7 line 3: The WRF model has been used with a 60 km horizontal resolution, while it is mentioned it is a non-hydrostatic model. Has the simulation really been carried out with non-hydrostatic physics ? It is surprising for such a resolution.

- Page 7 line 5: To explain this method of "spectral nudging", it would be useful to provide the approximate limit (in km) of the horizontal scales which are nudged towards NCEP analyses.

- Page 8 section 3.2.2: there is a confusion between POM (Primary or Particulate Organic Matter) and PPM. Please clarify the difference and correct the acronyms (between the text and Table 3).

- Page 9 lines 17-18: "all kind of anthropogenic and natural sources are taken into account on an hourly basis". Please clarify the time frequency of the meteorological forcing given by WRF, and the emissions in CHIMERE. Does this hourly frequency also concern natural emissions (sea-salt, dust) ?

- Page 10 Table 4: Please confirm that there is no coarse mode for sulfate. The text is misleading (line 8).

- Page 12 Section 4.1 and 4.2: how have the optical properties for the different aerosol bins been fixed ? This may explain partly the difficulties of CHIMERE to reproduce AOD and Angstrom exponent.

- Page 13 line 12: which aerosols could explain this "thin plume modeled over the Atlantic" ?

- Page 13 line 20: "high AE values" in western Europe. Contrary to what the authors affirm, this result is not found in the model (about 0.5 against values higher than 1 in the observations).

- Page 14 Section 5.1: In the PM10 series, are dust and sea-salt particles included ? Both for models and observations ? It is unclear for me, as these aerosols are not detailed in Section 6.

- Page 14 Section 5.2: The authors do not use the information on altitudes given in Table 1. As the comparison is done with first vertical level in the model, this could explain the difficulties of CHIMERE in the Alps for example.

- Page 17 line 13: K-puszta is not "the only station with a very poor correlation".

- Page 20 lines 3-5 (Section 7.1): I don't understand if this calculation is done on the model bins (as mentioned line 3) or on the AERONET bins (line 4), and in the second case, why the finest and coarsest sizes should be added, while they are not represented in Figure 9. Please clarify this method.

- Section 7.2: This diagnostic is original, and the authors draw several conclusions from Figure 11. However, it is not easy to understand. In particular, I suggest the authors to clarify the methodology presented in the beginning of this section and the caption in Figure 11.

- Page 23 line 14: please clarify "to integrate the dv/dlog(r)".

- Page 24 Figure 12: Please detail the caption.

———————————————————— Technical corrections:

- The line numbering should be corrected (several identical numbers in the same page).

- Please pay attention to the choice of American/English spelling, and keep it in the whole manuscript (for example modeled or modelled).

- Please remove all the articles "the" before dates.

- Abstract line 4: PM is not defined.

- Page 3 line 5: description (without s)

- Page 3 line 6: aerosol (without s)

- Page 3 line 10: please rephrase "the aerosol's composition behavior understanding"

- Page 3 line 27: Aerosol (without s)

- Page 5 Table 1 (caption): aerosol (without s)

- Page 7 line 18: concentration (without s)

- Page 9: 3.3.2 Emission distributions in aerosol bins

- Page 13 line 12: the Atlantic Ocean (and not sea)

- Page 23 line 20: underestimates

- Page 23 line 23: the

- Page 25 line 12: aerosol (without s)

———————————————————

---

## Author Comment (AC1) · 14 Sep 2016

**Review of paper acp-2016-275**
**Observations and regional modeling of aerosol speciation and size distribution over Africa and Europe, L. Menut et al.**

Dear Editor and reviewers,

We acknowledge the reviewers for the time spent to evaluate our work. We also acknowledge the Editor and we made all proposed changes in the revised manuscript.
Please note that our answers are in blue in the text and after each reviewers remark.

Best regards,
Laurent MENUT
September 14, 2016

**Answers to Anonymous Referee #1**

This manuscript, entitled "Observations and regional modeling of aerosol speciation and size distribution over Africa and Europe" presents an original analysis of aerosol properties, both modeled by a regional chemistry-transport model and given by surface and column-integrated observations, in terms of speciation and size distribution. Detailed results are about different aerosol species, from both natural and anthropogenic origins, are discussed, showing the performance of this regional model to reproduce the variety of aerosols in this region. The methodology and the results are for most of them (except in Section 7) clearly presented and explained, with discussion on the strengths and weaknesses of the current version of the model. However, several corrections are needed before considering a publication in ACP.
Thanks for this review and note that all remarks were taken into account and many parts of the article were changed accordingly. Detailed answers are after each remark.

**Main comments:**

1. Even if the comparisons between the model and observations show for different variables satisfactory scores for CHIMERE, the authors tend to overestimate its performance throughout the manuscript, notably in the following cases: - the correlations for AOD (Table 5, Section 4.1) are very weak for many stations (9 out of 28 have a correlation lower than 0.1). - the optical properties maps (Figure 3) show difficulties to reproduce correctly the Angstrom exponent in Western Europe - in the PM2.5 time series, some observed peaks are missed by the model, which is not explained (for example around 10 July in Schauinsland) I do not really challenge the performance of the model as I am aware of the complexity to compare observations and models, but rather the way it is presented by the authors (for example lines16-18 page 11). The latter should moderate their conclusions, and give possible reasons to explain the difficulties mentioned above.

   We understand this remark and the way to present the results was changed and moderated. For the 10 july, this is a typing error (see answer below) and this was corrected.

2. The title does not exactly correspond to the manuscript in terms of variables (Section 4 deals with aerosol optical properties which are not mentioned in the title) and of the domain of study ("Africa and Europe" is not precise enough). I suggest to add in the title optical properties, and to replace "Africa and Europe" by "Western Europe and Northern Africa".

   The title was extended as suggested by the reviewer for the domains. This is now: "*Observations and regional modeling of aerosol optical properties, speciation and size distribution over Northern Africa and Western Europe*".

3. The spatial distribution of the stations used here could be a major limitation of the study. Indeed, the EMEP stations cover only northern Europe and Spain (Africa and southern Europe are missing), while the AERONET stations do not include northern Europe. To my

knowledge, there are other stations which could be taken into account, which would improve the robustness of the different scores.

We completely understand this remark and the reviewer #2 also discusses this point. By definition, the EMEP stations are only in Europe. Some specific measurements were done in Islands during Adrimed for surface concentrations, but we showed in a previous study that the model resolution used for this work s not enough resolved to ensure correct comparisons: we need, with this large modeled domain, to use only 'background continental' stations. The use of 'coastal' stations will become possible only with an horizontal resolution less than ≈10 km with the model. And this is not the goal of this study being mainly dedicated to the aerosol life cycle during long-range transport. To correctly catch this long-range transport, it is needed to have a large domain and thus, due to computational realism, a low resolution. For the robustness of the scores: we think that the use of 46 EMEP stations and 28 AERONET stations is correct and representative.

4. The analyses on the speciation data are very interesting (notably Figures 5 and 6), but the same kind of diagnostics is missing as far as optical properties are concerned. I suggest to the authors to add this information on speciation for AOD, in order to better understand the differences shown in Figure 3.

This is difficult to make the same kind of diagnostic for aerosol properties. For the surface concentrations in mass, the speciation is easily obtained by splitting the total $PM_{2.5}$ or $PM_{10}$ with each chemical species. But for the optical properties, this is not possible. As many models CHIMERE calculates the optical properties using the 'external mixing' hypothesis. It is considered that the AOD is the sum of the AODs of each aerosol (each aerosol having its own optical properties, previously estimated using a Mie code). In the case of the CHIMERE model, this calculation is performed using the fastJX model, implemented on-line. The relative contributions are calculated inside FastJX but not externalized, only the resulting AOD is known. Thus, this is not possible to retrieve after the calculation the relative contribution of each aerosol in the AOD budget. But this is a good idea and probably a future development to achieve with the model.

**Minor comments:**

- Abstract lines 6-7: I don't understand why only mineral dust is mentioned in this sentence, while spatial correlation and daily variability concern all aerosols.
  Mineral dust is cited here as "main part of the long-range transport", but yes all aerosols are involved in the complete budget.
- Abstract line 12: "dust relative contribution": Please precise the variable (mass ?).
  Yes, this is the mass and this was added in the text. All units for aerosols are $\mu g.m^{-3}$ in the article.
- Page 3 line 18: what does the ADRIMED project bring to the present study ? Are there any specific observations that could be used here ?
  This study is a continuation of studies with the same modeling system and for the ADRIMED experiment. The direct comparison to specific ADRIMED measurements was already presented in [Menut et al., 2015a] and are thus not repeated here.
- Page 5 lines 17-18: The authors should justify properly that "the domain was selected to be large enough to account for anthropogenic emissions". The latitude of the northern limit of the domain seems to be too low, as northern British Isles, a part of northern Sea, and Scandinavia are not included in the domain.
  Yes, that's right. But the main goal of the study is to explore gas and aerosol concentrations around the Mediterranean sea. The sentence was changed to be more precise about this. The new sentence is: "*This domain was selected to be sure to have all sources producing gas and aerosol concentrations around the Mediterranean basin: European anthropogenic emissions, mineral dust and vegetation fires emissions.*".
- Page 7 line 1: "the results are presented from 10th June to 31st July 2013". Is it 30th (as in the following sections) or 31st July ?
  This is 30th and this was corrected in the text.

- Page7 line 3: The WRF model has been used with a 60 km horizontal resolution, while it is mentioned it is a non-hydrostatic model. Has the simulation really been carried out with non-hydrostatic physics ? It is surprising for such a resolution.
  This is difficult to really quantify the 'horizontal threshold' where an hydrostatic model remains correct and the threshold when a non-hydrostatic model becomes mandatory. With a spatial resolution of 60km and with a non-negligible topography (Alps, Atlas), we think this is preferable to use a non-hydrostatic model and to solve explicitely the pressure field. And even if this is not sure this is really useful, this is not false to really calculate the pressure instead of using the hydrostatic approximation.
- Page 7 line 5: To explain this method of "spectral nudging", it would be useful to provide the approximate limit (in km) of the horizontal scales which are nudged towards NCEP analyses.
  The choice to have wave numbers less than 3 corresponds to all wavelength greater than 2000km. This was added in the text.
- Page 8 section 3.2.2: there is a confusion between POM (Primary or Particulate Organic Matter) and PPM. Please clarify the difference and correct the acronyms (between the text and Table 3).
  Yes, this is right. PPM stands for "Primary Particulate Matter" (inorganic matter) and POM to "Primary Organic Matter". This was corrected accordingly in Table 3.
- Page 9 lines 17-18: "all kind of anthropogenic and natural sources are taken into account on an hourly basis". Please clarify the time frequency of the meteorological forcing given by WRF, and the emissions in CHIMERE. Does this hourly frequency also concern natural emissions (sea-salt, dust) ?
  More details were added in the text. At the beginning of the CHIMERE section, we added that CHIMERE used "hourly WRF meteorological fields". In the "emissions" section, this sentence was added: "In this model version, all kind of anthropogenic and natural sources are taken into account on an hourly basis: the anthropogenic emissions are estimated using hourly time profiles and are this hourly provided. The biogenic and mineral dust emissions (calculated on-line in CHIMERE) are using meteorological data and are also hourly estimated."
- Page 10 Table 4: Please confirm that there is no coarse mode for sulfate. The text is misleading (line 8).
  Yes, correct. There was an error in the text. The text is now "*For the anthropogenic emissions, the species POM, EC and PPM are emitted only in the fine and coarse mode, with MMMD of 0.2 µm and 4 µm, respectively. $SO_4$ is emitted in the fine mode only.*"
- Page 12 Section 4.1 and 4.2: how have the optical properties for the different aerosol bins been fixed ? This may explain partly the difficulties of CHIMERE to reproduce AOD and Angstrom exponent.
  Yes, this is right, this part is not detailed in this paper but is very important. We implemented a specific procedure and tested it a lot: we found a way to have the less uncertainty as possible. The important point is to have a stable AOD and AE calculation, independently of the number of bins of a model and of the size of each bin. The explanation is long but is completely explained in the [Mailler et al., 2016] article, the new reference paper for the CHIMERE model. This was added in the text in section 3.2.1.
- Page 13 line 12: which aerosols could explain this "thin plume modeled over the Atlantic" ?
  This is mineral dust. This could be see using two informations: (i) the plume comes from Africa, (ii) the Angstrom Exponent is low and characteristic of mineral dust. This was added in the text.
- Page 13 line 20: "high AE values" in western Europe. Contrary to what the authors affirm, this result is not found in the model (about 0.5 against values higher than 1 in the observations).
  Over Germany and Benelux, AE modeled values are up to 1. Probably, the reviewer wants we add details about the exact locations. Thus the sentence is now: "*The model is in good agreement with the measurements and the AOD values, between 0.1 and 0.5, are well located by the model. As for the 4 july, the regions composed by Germany and Benelux is mainly driven by high AE values, corresponding to more fine than coarse aerosol in the whole column: this result is both found for observations and model.*"
- Page 14 Section 5.1: In the PM10 series, are dust and sea-salt particles included? Both for models and observations? It is unclear for me, as these aerosols are not detailed in Section 6.

Yes, the explanation is not in section 6 but in section 3.2.2 and Table 3. A reminder is added at the beginning of section 5. This new sentence is: "This section is dedicated to the comparison between model and observations of $PM_{2.5}$ and $PM_{10}$. These "Particulate Matter" families correpond to the sum of all aerosols described in Table 3, for mean mass median diameter lower than $D_p$=2.5 $\mu$m and 10 $\mu$m, respectively."

- Page 14 Section 5.2: The authors do not use the information on altitudes given in Table 1. As the comparison is done with first vertical level in the model, this could explain the difficulties of CHIMERE in the Alps for example.
  Yes, that's right. The relative altitude of the site compared to the mean averaged altitude in a grid cell could change a lot the scores. For AOD this is not really a problem, this variable being vertically integrated in the whole atmospheric column. But, this is clearly a problem for surface concentrations. A new text was added about this limitation in section 2.1: *"The representativity of the station depends on the sub-grid scale variability of the model cell: more the variability is low more the station is representative. Over mountains areas, this is rare and, generally, stations at high altitude ASL can not be considered as well representative of the first model level for concentrations. In our case, this is probably the case for the stations in the Alps. In this study, these stations were considered for the scores calculations but, in case of poor comparisons scores, this problem of representativity could be a large part of the differences between model and observations. This is discussed in each case and in the following sections."*

- Page 17 line 13: K-puszta is not "the only station with a very poor correlation".
  Yes, this is a typing error and the sentence was corrected as: *"the only station with a negative correlation".*

- Page 20 lines 3-5 (Section 7.1): I don't understand if this calculation is done on the model bins (as mentioned line 3) or on the AERONET bins (line 4), and in the second case, why the finest and coarsest sizes should be added, while they are not represented in Figure 9. Please clarify this method.
  The text was changed to be more clear. The new sentences are now: *"As presented in section 2, the AERONET inversion products provide ASD for 15 bins, following a logarithmic distribution, ranging from 0.05 to 15 $\mu$m. In order to conserve all model information, the calculation is done on the AERONET bins plus extra bins in the finest and coarsest sizes: 5 bins are added below 0.05 $\mu$m with $r$=0.005, 0.01, 0.02, 0.03 and 0.04 $\mu$m and 3 bins are added after 15 $\mu$m with $r$=20, 30 and 40 $\mu$m. The model bins are interpolated on the AERONET bins and the column aerosol volume size distribution is calculated for each bin i as in (Pere et al., 2010):...".* Note also the corrected typo (AERONERT, p.20, l.5)

- Section 7.2: This diagnostic is original, and the authors draw several conclusions from Figure 11. However, it is not easy to understand. In particular, I suggest the authors to clarify the methodology presented in the beginning of this section and the caption in Figure 11.
  The diagnostic is just to sum the values of the concentrations before and fater the local minimum fixed with the value $r$=0.5 $\mu$m.

- Page 23 line 14: please clarify "to integrate the dv/dlog(r)".
  The text was probably hard to understand because the word "integrate" is not appropriate. In fact, this is just the cumul (sum) of the concentrations before and after the local minimum value. This was corrected in the text.

- Page 24 Figure 12: Please detail the caption.
  The caption was changed and is now: *"Scatter-plots for comparisons between the observations and the model for the aerosol size distribution. Each plot corresponds to the sum of the concentrations of aerosol for the "fine" mode (r ¡ 0.5 $\mu$m) and the "coarse" mode (r ¿ 0.5 $\mu$m). Each point corresponds to an hour during the whole simulation and a modeled concentration corresponding to an AERONET site. The sites are splitted in three families: Africa (black symbols), Europe (red symbols) and Mediterranean (green symbols), following the classification explained in Table 2."*

- Page 25 line 12: aerosol (without s)
  Ok corrected.

**Technical corrections:**

- The line numbering should be corrected (several identical numbers in the same page).
  The lines numbers are automatically generated by the Copernicus Latex package and this is not possible for us to change it.
- Please pay attention to the choice of American/English spelling, and keep it in the whole manuscript (for example modeled or modelled).
  Ok, the text was revised accordingly.
- Please remove all the articles "the" before dates.
  Ok, the text was revised accordingly.
- Abstract line 4: PM is not defined.
  "Particulate Matter" was added in the abstract.
- Page 3 line 5: description (without s)
  Ok corrected.
- Page 3 line 6: aerosol (without s)
  Ok corrected.
- Page 3 line 10: please rephrase "the aerosol's composition behavior understanding"
  OK. The sentence is now: *To go further in the aerosol life cycle understanding, it is now necessary...*
- Page 3 line 27: Aerosol (without s)
  Ok corrected.
- Page 5 Table 1 (caption): aerosol (without s)
  Ok corrected.
- Page 7 line 18: concentration (without s)
  Ok corrected.
- Page 9: 3.3.2 Emission distributions in aerosol bins
  Ok corrected.
- Page 13 line 12: the Atlantic Ocean (and not sea)
  Ok corrected.
- Page 23 line 20: underestimates
  Ok corrected.
- Page 23 line 23: the
  Ok corrected.
- Page 25 line 12: aerosol (without s)
  Ok corrected.

**Answers to Anonymous Referee #2**

This paper provides a thorough comparison of the performance of the CHIMERE model against a range of ground-based (EMEP stations) and column integrated (NASA-AERONET) aerosol observations and tests the capacity of the model to reproduce critical properties like chemical composition and size distribution over a vast region (Africa/Europe) with contrasted aerosol sources.
This paper provides a substantial contribution to scientific progress within the scope of ACP (substantial new concepts, ideas, methods, and data). The scientific approach and applied methods are correctly presented and the results discussed in an appropriate and balanced way. The scientific results and conclusions presented in a clear, concise, and well-structured way. I do recommend the publication of this paper after addressing the few minor comments reported below:

- Introduction: Can the authors better justify (in one or two sentences) why they have decided to focus on Europe/Mediterranean and why they have decided to use WRF + CHIMERE? Part of the explanation is provided at the beginning of the modelling section and could be moved in the introduction.
  The beginning of the modelling section was moved in the introduction and the new sentence is now: "To answer these questions, numerical simulations are performed for the two months of June and July 2013 and over a large domain encompassing Africa and Europe. This period corresponds to the ADRIMED project presented in [Mallet et al., 2016]. The simulations are

performed using two models: (i) the WRF meteorological model calculates the meteorological variables, (ii) the CHIMERE chemistry-transport model calculates the fields concentrations of gaseous and aerosols using the meteorological fields. WRF and CHIMERE are widely used for regional studies of atmospheric of gaseous and aerosol species. Over this domain and for this period, the two models were already used in [Menut et al., 2015a], [Menut et al., 2015b] and [**?**] and showed realitic results for the modeling of gaseous and aerosol species. In this study, the analysis is focused on the aerosol size distribution and its speciation in Africa and Europe."

- Page 2, line 10: "the size distribution modeling is poorly addressed in the literature". Can the authors justify in one sentence why it is a critical parameter that needs to be better constrained by model?
  Probably p.3, l.10. The sentence was rewritten as follows: "The chemistry of secondary organic species and deposition are also a source of uncertainties (Bergstrom et al., 2012; Fountoukis et al., 2014). More generally for aerosol, one part of this uncertainty is linked to the fact that the size distribution modeling is poorly adressed in the literature. This size distribution will directly impact the aerosol behaviour via the chemistry (nucleation, coagulation), the dry deposition (the settling velocity) and the wet deposition (the scavenging)."

- Figure 1: Why the selected NASA-AERONET stations are mostly located in Africa and Southern Europe and the EMEP stations mostly in the Northern Europe? There are also NASA-AERONET stations in Northern Europe and EMEP stations in Southern Europe (Italy, Greece, Cyprus ...)'
  This is a good remark and the choice of the stations was extensively discussed between all authors. In fact, this is mainly linked to the model resolution. In this study, in order to have only one modelled domain with all sources and transport pathways and, at the same time, in order to have enough computational resources to run several months with an hourly time-step and for all gaseous and aerosol species, we were constrained to have a 60kmx60km horizontal resolution. With this kind of resolution, the EMEP stations are mainly representative over land and when they are classified as 'background'. For AERONET, this problem does not appear, the photometers providing a vertically column-integrated measurements (and not a surface measurements as with EMEP). In conclusion, our selection of stations (and they are numerous) follows the "rules": EMEP for background and continental areas, AERONET for continental and/or islands and/or coastal areas. For future studies, we will probably optimize the CHIMERE model to be able to model such period with the same model characteristics but wwith a better horizontal resolution (10km): in this case, we will add all coastal stations (such as Italy, Greece and Cyprus). This will probably lead to interesting new questions and results about the aerosol chemical life-cycle.

- Page 5, line 18 : "... and transport of long-lived species across the Mediterranean Basin." What are these long-lived species?
  Details are added in the sentence as: "....and transport of long-lived species across the Mediterranean basin. These species are mainly ozone and CO for the gaseous species, mineral dust and organic matter (due to vegetation fires) for the aerosol."

- Table 2: Can the authors add a column "country" to help the reader to better locate the stations?
  Yes, this was done.

- Page 7, line 9 to 15 (section 3.1): An exhaustive description of the model is reported here. Is it the first time that all of these features are applied to CHIMERE? If not, can the authors provide some references which have used CHIMERE with these features?
  There is two model descriptions: the part highlighted by the reviewer is about the meteorological model, WRF. WRF is a "toolbox" and this is important to just cite the main parameterizations used. For CHIMERE, the main schemes are just referenced and the detailed description is only related to model choices important to well understand the results in the paper and not described before in publications. Some parts are in the model documentation but this is not considered as a peer-reviewed publication, so we have to add it in this article. However, a simplification was made here and the sea salt scheme of (Monahan, 1986) is just cited now.

- Table 3: I still don't understand what are the chemical species behind the term "primary particulate matter" (PPM). Can you be more specific?
  This is the mass of non-reactive aerosol provided in the HTAP emissions inventory. This was clarified in the text.

- Page 9, line 5 (section 3.3.2.). Can the authors provide a size range for fine/coarse/big?
  The size range is described in table 4: each mode is represented using a log-normal distributions with its mean mass median diameter and $\sigma$. The size range finally depends of the bins number. For our specific case of ten bins, the resulting size range is displayed in Figure 2. This is better explained in the text now.
- Table 4: What's about the other chemical species described in Table 3?
  The differences between Table 3 and Table 4 are for the species SOA, NO3 (nitrate) and NH4 (ammonium). The Table 3 describes the emitted species and the Table 4 the chemical aerosols taken into account in the model. The differences between the two tables are just for the aerosol species not directly emitted but considered for the chemistry in the model.
- Figure 2: This figure provides the % distribution in the different modes for the different chemical species. Since this figure is scaled with relative contributions (%), it is not possible, for instance, to learn what is the fine versus coarse contribution of SO4 (e.g. what is the contribution of fine SO4 relatively to total SO4)?
  Yes, this figure represents only the emitted species. In our inventory, only the fine fraction of SO4 is emitted. This is why only this distribution is plotted. For the complete size distribution of SO4, this will depends after on chemistry, transport, deposition and there is thus no constant ratio of fine/total for this species.
- Table 5: Can the authors specify what the rationale of the classification of the stations is? (e.g. why this order?).
  The order is in increasing latitude. This explanation was missing and is now in the caption. This enables to have the same order than for the Figure 6. This is true for all results Tables.
- Page 13, line 17: Can the authors explain why they have selected specifically these 3 days?
  Yes, we agree this is not explained and not clear. These figures were selected after screening all days of the period. They were selected for three reasons: (i) the studied plumes are well defined, clearly identified, and the message for the analysis is more clear with these days. (ii) Aeronet data are not always available (depending the the day) and these three days correspond to a large amount of data to superimpose on the map. (iii) The first day (18 june) corresponds to a strong peak discussed in the article (section 5.2). The two other days are, more or less, with a step of two weeks, leading to a correct temporal coverage for the discussion. These reasons were added in the text.
- Page 17, line12: "... the important peak of PM2.5 observed around the 18 July ...". I cannot see clearly a peak of PM2.5 on 18.07 in Figure 5. Can the authors better point this peak?
  p.14, l.18, no? and yes, sorry, this is the peak of 18 june (and not july). This was corrected in the text.
- Page 19, end of the page, discussion related to NH4. The authors discuss on the model performance for NH4 taken individually while this compound is intimately linked to fine SO4 (and fine NO3 if there is). How the neutralization of SO4 by NH4 is reproduced by the model?
  Yes, NH4 is directly linked to fine SO4 and the two are discussed. In the text, p.20 l.15, we noted: *"Performances on ammonium follow the ones of sulfate, most of the ammonium reacts with sulfuric acid to form ammonium sulfate salts."* This process is performed using the ISOROPIA model for the thermodynamical equilibrium, [Nenes et al., 1998], implemented on-line in CHIMERE and with the parameterization of [Kulmala et al., 1998] for the sulfuric acid nucleation. These details are included in the CHIMERE model presentation article, [Menut et al., 2013].
- Page 22, line 4. Although we usually find the highest concentrations of fine NO3 close to the English channels, I would assume that it is rather due to thermodynamic equilibrium (cold and humid) rather than high NOx from shipping emissions. Can the authors demonstrate that fine NO3 in this region is mostly due to shipping rather than NOx for the continental (transport) sector? Otherwise I would remain vague about the origin of NOx in the region.
  We assume this remark is related to the sentence on page 19 and line 9. Correct? Whatever, we agree with the reviewer remark: we can not prove that this NO3 is majoritarily due to shipping emissions in the Channel. This could be due to anthropogenic emissions. Two reasons to be careful: (i) the model resolution is 60km, sometimes larger than the Channel width. Thus, anthropegenic and shipping emissions are colocated in the same model grid cell. This is not possible to separate the two contributions with this model resolution. The sentence was changed and is now: *The addition of $NO_x$ shipping and anthropogenic emissions (advected above the sea)*

*is responsible for the formation of nitrate favored by mild, humid conditions and low deposition over the Channel.*

- Figure 9: Why the authors did not take daily averages here (which are more representative than a given hour)?
  For this Figure, the hourly representation was chosen and this may appear strange since all results are presented as daily averaged values in the paper. But in this specific case, we are superimposing the AERONET size distribution inversion product and there is no data for all hours. A choice could be to average the model exactly with the hour available in the data (sometimes there is only one or two hours per day) but this means that we will have some "daily averaged" plots made with one or two hours and some others with 24 hours. To avoid such additional uncertainty, we chosen to present hourly size distributions.

**References**

[Kulmala et al., 1998] Kulmala, M., A., L., and Pirjola, L. (1998). Parameterization for sulfuric acid / water nucleation rates. *J. Geophys. Res.*, 103(No D7):8301–8307.

[Mailler et al., 2016] Mailler, S., Menut, L., Khvorostyanov, D., Valari, M., Couvidat, F., Siour, G., Turquety, S., Briant, R., Tuccella, P., Bessagnet, B., Colette, A., Létinois, L., and Meleux, F. (2016). Chimere-2016: From urban to hemispheric chemistry-transport modeling. *Geoscientific Model Development Discussions*, 2016:1–41.

[Mallet et al., 2016] Mallet, M., Dulac, F., Formenti, P., Nabat, P., Sciare, J., Roberts, G., Pelon, J., Ancellet, G., Tanré, D., Parol, F., Denjean, C., Brogniez, G., di Sarra, A., Alados-Arboledas, L., Arndt, J., Auriol, F., Blarel, L., Bourrianne, T., Chazette, P., Chevaillier, S., Claeys, M., D'Anna, B., Derimian, Y., Desboeufs, K., Di Iorio, T., Doussin, J.-F., Durand, P., Féron, A., Freney, E., Gaimoz, C., Goloub, P., Gómez-Amo, J. L., Granados-Munoz, M. J., Grand, N., Hamonou, E., Jankowiak, I., Jeannot, M., Léon, J.-F., Maillé, M., Mailler, S., Meloni, D., Menut, L., Momboisse, G., Nicolas, J., Podvin, T., Pont, V., Rea, G., Renard, J.-B., Roblou, L., Schepanski, K., Schwarzenboeck, A., Sellegri, K., Sicard, M., Solmon, F., Somot, S., Torres, B., Totems, J., Triquet, S., Verdier, N., Verwaerde, C., Waquet, F., Wenger, J., and Zapf, P. (2016). Overview of the chemistry-aerosol mediterranean experiment/aerosol direct radiative forcing on the mediterranean climate (charmex/adrimed) summer 2013 campaign. *Atmospheric Chemistry and Physics*, 16(2):455–504.

[Menut et al., 2013] Menut, L., Bessagnet, B., Khvorostyanov, D., Beekmann, M., Blond, N., Colette, A., Coll, I., Curci, G., Foret, F., Hodzic, A., Mailler, S., Meleux, F., Monge, J., Pison, I., Siour, G., Turquety, S., Valari, M., Vautard, R., and Vivanco, M. (2013). CHIMERE 2013: a model for regional atmospheric composition modelling. *Geoscientific Model Development*, 6:981–1028.

[Menut et al., 2015a] Menut, L., Mailler, S., Siour, G., Bessagnet, B., Turquety, S., Rea, G., Briant, R., Mallet, M., Sciare, J., Formenti, P., and Meleux, F. (2015a). Ozone and aerosol tropospheric concentrations variability analyzed using the ADRIMED measurements and the WRF and CHIMERE models. *Atmospheric Chemistry and Physics*, 15(11):6159–6182.

[Menut et al., 2015b] Menut, L., Rea, G., Mailler, S., Khvorostyanov, D., and Turquety, S. (2015b). Aerosol forecast over the Mediterranean area during July 2013 (ADRIMED/CHARMEX). *Atmospheric Chemistry and Physics*, 15(14):7897–7911.

[Nenes et al., 1998] Nenes, A., Pilinis, C., and Pandis, S. (1998). ISORROPIA: A new thermodynamic model for inorganic multicomponent atmospheric aerosols. *Aquatic Geochem.*, 4:123–152.